



# Potential application of hydrological ensemble prediction in forecasting flood and its components over the Yarlung Zangbo River Basin, China

Li Liu[1], Su L. Pan[1], Zhi X. Bai[1], Yue P. Xu[1]

[1]Institute of Hydrology and Water Resources, Civil Engineering and Architecture, Zhejiang University, Hangzhou, 310058, China

*Correspondence to*: Yue P. Xu (yuepingxu@zju.edu.cn)

**Abstract.** In recent year, flood becomes a serious issue in Tibetan Plateau (TP) due to climate change. Many studies have shown that ensemble flood forecasting based on numerical weather predictions can provide early warning with extended lead time. However, the role of hydrological ensemble prediction in forecasting flood volume and its components over the Yarlung Zangbo River Basin (YZR), China has not been systematically investigated. This study adopts Variable Infiltration Capacity (VIC) model to forecast annual maximum floods (MF) and annual first floods (FF) in YZR based on precipitation, maximum and minimum temperature from European Centre for Medium-Range Weather Forecasts (ECMWF). N-simulations is proposed to account for more scenarios of parameters in VIC and shows improved flood simulation. Ensemble flood forecasting system can skilfully predict MF with a lead time of more than10 days, and has skill in forecasting the snowmelt-related components in about 7 days ahead. The accuracy of forecasts for FF is inferior with a lead time of only 5 days. The performance in 7-day accumulated flood volumes is better than the peak flows. The components in baseflow for FF are irrelevant to lead time, whilst for MF an obvious deterioration in performance with lead time can be perceived. The snowmelt-induced surface runoff is the most poorly captured component by the system, and the well-predicted rainfall-related components are the major contributor for good performance. From this study, it is concluded that snowmelt-induced flood volume plays an important role in YZR Basin especially in FF.

## 1. Introduction

The Tibetan Plateau (TP) as the source of many major rivers is known as "the world water tower" (Xu et al., 2008). Due to the special geological, topographic and meteorological conditions, the ecosystem in this area is vulnerable and susceptible to climate changes (Zhao et al., 2006). According to previous researches, it is confirmed that the atmospheric and hydrological cycle in TP have underwent significant changes. Evident climate warming (Guo and Wang, 2012; Wang et al., 2014; Yang et al., 2014), increased precipitation (Kuang and Jiao, 2016, Wang et al., 2017), glacier retreat and permafrost degradation (Cheng and Wu, 2007) can be perceived, and these impacts are expected to be exacerbated by future climate change (Su et al., 2013). As a result, frequent natural disasters such as flooding and debris flow take place with an estimated direct economic loss amounting to 100 million RMB per year (Zhang et al., 2001). Thus, seeking advanced techniques to improve the accuracy of flood forecasts plays a critical important role in enhancing disaster resilience (Kalra et al., 2013; Yucel et al., 2015; Girons et al., 2017).



It is now a routine practice to introduce the Numerical Weather Prediction (NWP) products into operational and research flood forecasting system to generate ensemble streamflow forecasts (Cloke and Pappenberger, 2009). Compared with traditional single-value deterministic flood forecasts, it has been verified that the forecasts based on Hydrological Ensemble Prediction System (HEPS) outperform the traditional deterministic ones with higher accuracy and longer lead time

(Bartholmes et al., 2009; Cloke et al., 2013; 2017; Li et al., 2017; Pappenberger et al., 2015; Todini, 2017). Flood forecasting is one of the most important topics applying HEPS (Arheimer et al., 2011; Shi et al., 2015), but most of the studies only focus on peak flows (Alvarez-Garreton et al., 2015; Valeriano et al., 2010; Dittmann et al., 2009), and seldom studies have investigated the ability of HEPS forecasts in typical accumulated flood volumes and the respective components contributed to the flood volumes, especially the snowmelt-induced component. It is shown that snow water availability and snow

dynamics are issues of fundamental importance in high mountain hydrology (Bavera et al., 2012). Investigating the components constituting the total runoff facilitates the understanding of runoff generation mechanism and furtherly improving flood forecasting in high mountains where our study area is located.

Investigating the skill of HEPS in streamflow components requests effective methods to separate total runoff into different components of interest. Numerous researchers have studied the methods to achieve hydrograph separation. Some researchers

are interested in separating baseflow or groundwater component from total runoff. For example, Partington et al. (2011) developed a hydraulic mixing-cell method to determine the groundwater component and Luo et al. (2012) utilized the digital filter program to separate baseflow from streamflow. However, many of the hydrological models per se have the ability to separate streamflow into baseflow and surface runoff, like SWAT (Luo et al., 2012) and VIC model (Liang et al., 1994), thus the separation of snow/glacier-driven component gains increasing interests. The most common and historical practice to

separate snowmelt and glaciermelt components is to conduct stable isotope analysis (Isotopic hydrograph separation, HIS) (Laudon et al., 2002). Sun et al. (2016) applied HIS in the Aksu River and successfully calculated the relative contribution of the glacier and snow meltwater to total runoff. Besides the experimental approaches, considerable studies obtain snowmelt component via a simple ratio of rainfall and snowmelt from hydrological model simulation (Cuo et al., 2013a; Siderius et al., 2013), whereas these methods are often primitive and neglect the physical processes that affect the transformation from snow

to runoff, such as evapotranspiration, sublimation, and infiltration. Li et al. (2017) developed a new snowmelt tracking algorithm in VIC model to compute the ratio of the snow-derived runoff to the total runoff with consideration of systematic analyses, demonstrating promising performance in applications over western United States.

Generally, the successful application of hydrologic models depends on how well the models are calibrated. Yapo et al. (1998) showed there is no single objective function that can represent all the features of runoff hydrographs such as time to peak,

peak flow and runoff volume. Increasing investigators have realized that multiple objectives optimization can bring out better results than single ones, and currently majority of the hydrological models are calibrated based on multiobjective optimization algorithms (Kamali et al., 2013; Troy et al., 2008; Voisin et al., 2011; Yuan et al., 2013). Multiobjective formulation will result in a set of Pareto optimal solutions that represent trade-offs among different objectives (Wöhling et al., 2013). Thus, compromise is necessary (Gong et al., 2015). Most of the studies eventually select only one value from the



Pareto front to represent the model parameter set for their simulation (Troy et al., 2008; Voisin et al., 2011; Yuan et al., 2013; Liu et al., 2017). This value is usually the compromise point that balances the diverse and sometimes conflicting requirements. However, these solutions provided by multiobjective optimization algorithms have the property that moving from one to another along the tradeoff surface results in the improvement of one objective while causing deterioration in at

least one other objective. In some cases, as mentioned by Kollat et al. (2012), it is difficult to cause the two-objective trade-off to collapse to one single point. Due to this limitation, a few studies tend to utilize an ensemble of parameter sets to cover more probabilities in hydrological model state. Wöhling and Vrugt (2008) employed Bayesian model averaging to generate forecast ensembles of soil hydraulic models, showing as similar skills as the best ones. Teutschbein and Seibert (2012) employed 100 different optimized parameter sets in HBV to simulate streamflow with a wide range of potential sources of

variability. The basic principle in ensemble forecasts is using ensemble spread to quantify forecast uncertainty and thus provide essential information to users (Bauer et al., 2015). Analogous to this concept, the benefit of adopting an ensemble of parameter sets of hydrological model for flood and components forecasting to consider more possible hydrological initial conditions remains an unresolved and noteworthy question.

The two purposes of this study are therefore to investigate the forecast ability of HEPS in flood volume and its components

over mountainous area, and the impact of an ensemble of pareto optimal solutions on model simulations. To this end, the paper is structured as follows: Section 2 describes the information of study area and data used. Methodology description is in Section 3. Section 4 provides the result analysis, and Section 5 discusses the findings and points to future research directions, and conclusion is presented in Section 6.

## 2. Study area and data

### 2.1 Study area

We focused our analysis on the Yarlung Zangbo River Basin (YZR), located at the upper reaches of Brahmaputra River basin, which stretches across the southern part of TP from the west to the east, with a drainage area of $2.1 \times 10^5 \ km^2$ controlled by Nuxia hydrological station in China. The basin is selected for the greatest population density in TP, increasing runoff and glacier snow melt (Wang et al., 2009; Liu et al., 2014), making it an ideal region to investigate flood forecasting

and its components. YZR is one of the highest great rivers in the world with a mean elevation exceeding $4000 \ m$ a.s.l. The climate from upstream to downstream regions of the basin exhibits an obvious difference due to the location and the topographical feature of the TP (Liu et al., 2014). The downstream area has a warm and humid subtropical climate; the midstream area has a temperate forest-grassland climate and the upstream valley has a cold and dry temperate steppe climate (Liu et al., 2007; Shen et al., 2012). The average annual temperature in this basin is about 6.27 ℃. The average annual

precipitation is about $560 \ mm$, most of which occurs during wet season from May to September (Li et al., 2014). Approximately 1/3 of the basin area is covered by snow and glacier, resulting in a significant glacier-snow melt induced floods in late spring and early summer.

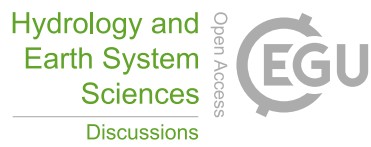

[Figure 1]

## 2.2 Data

The gauged meteorological data, including daily precipitation, minimum/ maximum temperature, wind speed and relative humidity, from 1998 to 2015 were collected from 27 National Meteorological Observatory stations located in and around the

YZR as shown in Fig.1. Daily streamflow from three hydrological stations were utilized in this study, i.e. Nuges Station (NGS), Yangcun Station (YC) and Nuxia Station (NX) from the most upstream to downstream region. Except data missing in 2009, the record period of observed streamflow at NGS and NX is consistent with that of the meteorological data and the period of observed streamflow at YC is shorter, spanning from 1998 to 2012. The first year was used as warm-up period. Periods from 1999 to 2005, 2006 to 2008 and 2010 to 2012/2015 were adopted for calibration, validation and evaluation

purpose respectively.

The daily Quantitative Precipitation Forecasts (QPF) and Maximum/Minimum Temperature (MXT/MNT) from 2007 to 2015 were obtained from European Centre for Medium-Range Weather Forecasts (ECMWF) with lead time from 24h to 360h. To be consistent with the observations, the data issued at 0000 UTC was downloaded. ECMWF was selected in this study as it is well-known that the forecasts from ECMWF are more skillful than other Ensemble Prediction Systems in

TIGGE database (Buizza et al., 2005; Froude, 2010; Tao et al., 2014). The additional data used for hydrological modeling is similar to our previous work (Liu et al., 2017), thus omitted in this paper.

## 3. Methodology

### 3.1 Hydrological model and N-Pareto-optimal parameter sets

The Variable Infiltration Capacity model (VIC, Liang et al., 1994; 1996) was employed in this study to investigate the ability

of ensemble flood forecasting in YZR. VIC is a well-established and extensively used rainfall-runoff model especially in areas with existence of snowmelt and frozen soil (Tang and Lettenmaier, 2010; Cuo et al., 2013a; Su et al., 2016). A two-layer snow model is embodied in VIC, which considers snow accumulation and ablation in a ground pack and an overlying forest canopy based on energy balance (Andreadis et al., 2009). The frozen soil algorithm makes it possible to represent the effects of seasonally frozen ground on surface water and energy fluxes (Cherkauer and Lettenmaier, 1999; 2003). These are

two of the critical elements in VIC that are particularly relevant to our research.

In this study, VIC was operated at a six-hourly time step in both water and energy balance model with a spatial resolution of $0.125° \times 0.125°$. The snow and frozen soil algorithms were active. Gauged and forecasted meteorological data were interpolated into the requested resolution using the Inverse Distance Weighted (IDW) method coupled with an elevation-based lapse rate. The lapse rate in this study was set as $0.6\ mm\ km^{-1}$ for precipitation and $-6.5\ °C\ km^{-1}$ for temperature.

These two values were determined by a cross-validation process and were roughly consistent with the findings in Cuo et al.



(2013b) which performed the least squares fitting on daily temperature and precipitation over the Northern TP to gain the best lapse rate for interpolation.

Model calibration was conducted by a parallel-programmed Epsilon-Dominance Non-Dominated Sorted Genetic Algorithm II (ε-NSGA II) as proposed by the authors (Liu et al. 2017). The ε-NSGA II was coupled with Message Passing Interface (MPI) to achieve parallel autocalibration with high efficiency. As the flood peak and volume are our focus in this study, more priorities are given on high flows when calibrating the model. Four objective functions were used for model calibration at three hydrological stations: the Nash–Sutcliffe efficiency and relative bias for all streamflow and for the top 10% streamflow. Detailed formulas are defined as:

$$NSE = 1 - \frac{\sum_{i=1}^{N}(Q_{obs}(i) - Q_{sim}(i))^2}{\sum_{i=1}^{N}(Q_{obs}(i) - \overline{Q_{obs}(i)})^2} \qquad (1)$$

$$Bias = \frac{\sum_{i=1}^{N}[Q_{sim}(i) - Q_{obs}(i)]}{\sum_{i=1}^{N}Q_{obs}(i)} 100\% \qquad (2)$$

$$NSE_{10\%} = 1 - \frac{\sum_{i=1}^{M}(Q_{obs,10\%}(i) - Q_{sim,10\%}(i))^2}{\sum_{i=1}^{M}(Q_{obs,10\%}(i) - \overline{Q_{obs,10\%}(i)})^2} \qquad (3)$$

$$Bias_{10\%} = \frac{\sum_{i=1}^{M}[Q_{sim,10\%}(i) - Q_{obs,10\%}(i)]}{\sum_{i=1}^{M}Q_{obs,10\%}(i)} 100\% \qquad (4)$$

in which $N$ and $M$ are the number of the daily and top 10% flows, respectively; $Q_{obs}$ and $Q_{sim}$ are the observed and simulated daily streamflow; and $Q_{obs,10\%}$ and $Q_{sim,10\%}$ are the observed and corresponding simulated top 10% flows, respectively.

After calibration, the N-Pareto-optimal parameter sets were selected according to the method of Preference Ordering Routine (POR) developed by Khu (2005). There are two key theorems for this method. The first is the efficiency of $k$ order (or $k$-Pareto-optimal points). Considering all the possible $k$-dimensional subspace of the original $m$-dimensional objective functions provided by ε-NSGA II ($1 \leq k \leq m, m = 4$ in this study), a point is defined as being efficient of order $k$ if this point is not dominated by any other points in any of the k-dimensional subspaces. The second theorem is the efficiency of order $k$ with degree $p$ (or $[k, p]$-Pareto-optimal points). A point is defined as being efficient of order $k$ with degree $p$ if it is not dominated by any other points for exactly $p$ out of the possible k-dimensional subspaces. By reducing the efficiency of order $k$ and increasing the degree of order $p$ in a sequential manner, POR is able to achieve the reduction of the number of possible solutions and to short-list the most relevant ones for retention as calibration parameters. Detailed procedures and examples to apply POR are omitted here, and interested readers can refer to Khu (2005).

In this study, the POR is performed until less than 10 points are obtained. Additionally, some other points on the Pareto front are also retained: the extreme value for each objective function (indicated by filled circles in Fig. 2) and the compromise value in the two-objective trade-off (indicated by filled star in Fig. 2). In this way, limited number of parameter sets is picked out to represent different scenarios of model state. For convenience, the simulations driven by the N-Pareto-optimal parameter sets are referred as N-simulations, and the simulation by only one parameter set (the compromise point) is indicated by S-simulation thereafter.





**3.2 Hydrograph separation**

The Snowmelt Tracking Algorithm (STA) proposed by Li et al. (2017) is adopted in this study to separate the hydrograph. Due to the lack of glacier algorithm in the VIC model, in this study, the glaciermelt is considered together with snowmelt, and the term of "snowmelt" is used for representation. Based on VIC modeling, in order to obtain the streamflow derived from snowmelt in total runoff $Q_{snow,t}$, STA calculates the snowmelt-induced streamflow in surface runoff ($R$) and baseflow ($B$) separately. For surface runoff derived from snowmelt ($R_{snow,t}$), STA assumes snowmelt and rainfall exhibit identical infiltration ($f_{i,snow,t}$) and runoff ($f_{R,snow,t}$) ratios. In this way, $R_{snow,t}$ is computed as a function of the ratio of snowmelt $M_t$ to snowmelt + rainfall, $M_t + Rain_t$:

$$R_{snow,t} = R_t f_{R,snow,t} = R_t f_{i,snow,t} = R_t \frac{M_t}{M_t + Rain_t} \qquad (5)$$

The fraction of baseflow induced by snowmelt ($f_{B,snow,t}$) is assumed to be equal to the proportion of soil moisture that originated as snowmelt in all soil moisture layers ($f_{W,snow,t}$). Thus:

$$B_{snow,t} = B_t f_{W,snow,t} \qquad (6)$$

Then, $f_{W,snow,t}$ is obtained by an iteration process. The formula used to obtain $f_{W,snow,t}$ is defined as follows:

$$f_{W,snow,t} W_t = f_{W,snow,t-1} W_{t-1} + f_{i,snow,t-1} i_t \Delta t - f_{W,snow,t-1}(ET_t - Sub_t)\Delta t - f_{W,snow,t-1} B_t \Delta t \qquad (7)$$

where $W_t$ and $ET_t$ are soil moisture and evapotranspiration output from VIC, respectively. Sublimation $Sub_t$ is calculated from the evolution of the snow water equivalent (SWE) on the ground.

A similar equation to Eq. (7) can be written for rain ($f_{W,rain,t}$). At each time step, $f_{W,snow,t} + f_{W,rain,t} + f_{W,unkown,t} = 1$. At step time $t = 1$, $f_{W,unkown,t} = 1$, indicating that the source of runoff (snowmelt or rainfall) is unknown at initial time step. After the tracking system performed, $f_{W,unkown,t}$ decreases to 0, and sum of $f_{W,snow,t}$ and $f_{W,rain,t}$ is equal to 1 with fully explained soil moisture sources.

Unlike Li et al. (2017), all the aforementioned variables are integrated values over the entire watershed. In this way, the streamflow is separated into four components, the surface runoff derived from snowmelt ($R_{snow,t}$) and from rainfall ($R_{rain,t}$); the baseflow derived from snowmelt ($B_{snow,t}$) and from rainfall ($B_{rain,t}$).

**3.3 Post-processing of forecasts from ECMWF**

In order to improve the raw forecasts from ECMWF, we proposed a post-processing method by coupling parameterized Quantile Mapping (QM) with Schaake Shuffle. QM is adopted in this study for it is a simple yet effective statistical bias correction method in hydrological applications (Li et al., 2010; Xu et al., 2014; Salathé et al., 2015). In most cases, the empirical cumulative distribution function is used to present the data distribution in QM. However, many studies (Viste et al., 2013; Stauffer et al., 2017; Tao et al., 2014) have demonstrated that it is more appropriate to use fitted parametric distributions as no frequent interpolation or extrapolation would be requested (Li et al., 2010). For QPFs, due to the strongly positively skewed distribution in rainfall (Stauffer et al., 2017), QM based on single gamma distribution was recommended





and utilized for bias correction, although some studies found that a combination of double-gamma (Yang et al., 2010) or gamma-GEV distribution (Smith et al., 2014) can be more effective. There were two reasons for our choice here. Firstly, we compared the single gamma with double gamma and gamma-GEV distribution, and obtained almost similar performance scores according to Mean Squared Error. Secondly, the bias correction in this study was performed for each grid, each lead

time and each variable. Considering the heavy computation labor, the single gamma distribution was selected for timesaving and efficiency. For MXT and MNT, four -parameter beta distribution was utilized as suggested by Li et al. (2010). Owing to the limited record of ECMWF forecast, the data from 2007 to 2009 was used as training data to determine the parameters for distribution.

As mentioned before, the forecasts for different lead time, grids and variables were post-processed independently. The

forecast ensembles therefore do not have appropriate space–time correlations. To generate ensemble members with appropriate space–time correlations, Schaake shuffle (Clark et al., 2004) was applied to link historical data to ensemble members and create sequences with realistic temporal-spatial patterns. 38 years of historical data from 1978 is used to apply the Schaake Shuffle procedure. Details to conduct Schaake Shuffle can be found in Clark et al. (2004) and Schepen et al. (2017).

## 3.4 Evaluation indicators

The annual maximum flood (MF) is picked out as typical flood events. Meanwhile, the first flooding (FF) event in each year is also selected. MF is determined by one-day peak flow, while for FF the definition seems to be kind of subjective. Nevertheless, FF is just introduced as an example to verify the skill of VIC/ECMWF system to predict the snowmelt. From this point of view, it doesn't matter whether it is the exact and objective first flood or not. There are four criterions for us to

define FF: (1) the peak flow should be more than twice of the average daily streamflow during dry period (November to March); (2) the duration for the flood event should be longer than 7 days; (3) the observed snowpack is present; (4) more than half of the N-Pareto-Optimal sets simulate the occurrence of snowmelt. Considering that MF flood events in YZR usually last for several months, flood volume over the entire flood event is impossible to be covered by medium-range weather forecasts. Four typical flood volumes are therefore chosen to represent the volume indexes, i.e. the peak flow (Q1),

the accumulative 3-days flows centered on peak flow (Q3), the accumulative 5-days flows centered on peak flow(Q5), the accumulative 7-days flows centered on peak flow (Q7). Term "duration" is adopted to represent the number of days used to generate flood volumes.

The Continuous Ranked Probability Skill Score (CRPSS) (Hersbach, 2000) is adopted to indicate the overall performance of the forecasts as a comprehensive evaluation metric, which is calculated via normalizing the Continuous Ranked Probability

Score (CRPS) by a reference forecast. The reference forecast in this study is an ensemble of hydrological forecasts simulated by the VIC model using sampled historical meteorological observations at the same calendar day as input to the model (Bennett et al., 2014). For deterministic forecasts, the CRPS reduces to Mean Absolute Error (MAE), and can be directly



compared. CRPS and MAE are negatively oriented and tend to increase with forecasts bias or poor reliability (Shrestha et al., 2015). The value of CRPSS ranges from -∞ to 1, with best score equal to 1.

Two specialized indicators for flood events are utilized according to research works by Smith et al. (2004), i.e., the percent absolute flood volume error $E_q$ and percent absolute peak time error $E_t$. The definitions are in formulas (8) - (9):

$E_q = \frac{\sum_{i=1}^{N}|B_i|}{NY_{avg}} \times 100$        (8)

$E_t = \frac{\sum_{i=1}^{N}|T_{pi}-T_{psi}|}{N} \times 100$      (9)

herein $B_i$ is the volume bias for $ith$ flood event; $Y_{avg}$ is the average observed flood volume for $N$ selected flood events. $T_{pi}$ and $T_{psi}$ are the observed and simulated time to $ith$ peak.

Additionally, two types of verification data are utilized in this study to compute the aforementioned statistical indicators if
applicable. One is the observation derived from hydrological station measurement, and the other is the simulation outputs from VIC driven by observed meteorological data. Indicators verified on observations show errors from both hydrological model and meteorological forecasts, whereas these verified on simulation outputs show errors only from meteorological forecasts.

## 4. Results

### 4.1 Hydrological model performance

Fig. 2 shows an example of two-dimensional Pareto plots for Bias and NSE at NGS. The performance of the selected N-Pareto-optimal parameter sets and traditional parameter set during calibration and evaluation periods for three hydrological stations are listed in Table 1. Broadly speaking, the model performance during evaluation is more satisfactory than that during calibration. It is probably caused by the existence of considerable extraordinary floods during the calibration period.
It is noticeable that simulation at NGS is better than that at other two stations with NSE greater than 0.77 for daily streamflow and NSE near 0.5 for top 10% streamflow. Performance at NX is inferior with bias greater than 30%, which is similar as the previous studies by Tong et al. (2014) and Zhang et al. (2012). They claimed that the underestimation in streamflow simulation at NX is highly likely to be caused by the largely underestimated CMA observations in this area, and we guess it is also a reason that within downstream regions the hydrological process becomes more complicated due to
human activities. The bolded values in the table are the cases that S-simulation behaves better than N-simulations according to the selected objective functions. It is obvious that in these cases S-simulation generally performs well during calibration, whilst during evaluation period it loses the advantage to some degree. The NSE of top 10% streamflow at NGS is the only one case where S-simulation consistently outperforms the N-simulations in either calibration period or evaluation period.
[Figure 2]





The observed and simulated hydrographs during evaluation at NGS are presented in Fig. 3. An obvious underestimation can be observed in low flow period. There are two possible reasons for this phenomenon. Firstly, the objective functions used for calibration have the tendency to give more attention to high flows as the flood is the focus of our investigation. Secondly, the absence of glacier module in VIC is believed to deteriorate model performance in some way. As noticed in Fig. 3, the flood peaks are well captured by S-simulation in most cases. N-simulations are able to cover all the extreme values while sometimes slight overestimation exists.

[Figure 3]

The performance indices of typical flood volumes simulated by VIC for FF and MF during the whole study period are listed in Table 2. Two statistical indictors are adopted here, CRPS for N-simulations and MAE for S-simulation. For NGS and YC, CRPS is consistently smaller than MAE, implying better simulation by N-simulations. On the contrary, S-simulation at NX consistently provides better performance than N-simulations, for the selected single parameter set at this station is actually the best parameter set for three of the objective functions, which can be viewed in Table 1.

[Table 1] [Table 2]

In order to present more detailed performance of flood volume simulation by VIC, Figure 4 exhibits typical flood volumes for MF from 2010 to 2015 respectively. It is noticeable that the majority of the flood events can be captured by N-simulations, and volumes tend to be better covered as with the "duration" increasing. The flood volume at YC is better simulated than that at the other two stations. It is consistent to the highest NSE for top 10% streamflow at this station as shown in Table 1. The floods at NX are obviously underestimated. In most of cases, the N-simulations even fail to cover the observations. Similar but better behaviors exist for FF and thus omit here.

[Figure 4]

VIC simulated snow cover was compared with snow depth derived from passive microwave remote-sensing data by Che et al. (2008) and Dai et al. (2015). Figure 5 shows the spatial distribution of observed and simulated daily average snow depths during evaluation. For simplicity, only the results at NX is displayed. An acceptable agreement can be found over the entire domain, especially for the middle reaches. Some overestimation exists in the upstream and downstream regions. Explanation for these errors in snow depth will be furtherly depicted in Section 5. We also compared the fraction of snowmelt-induced components to total runoff with previous studies (Liu, 1999; Cuo et al., 2014) as shown in Table 3. It is noticeable that the results by S-simulation are quite close to the records, except YC with 5% overestimation. Most of the records are covered by N-simulations, and at NX, all the parameter sets underestimate the snowmelt streamflow.

[Figure 5]  [Table 3]

## 4.2 Flood volumes forecasts

Lead times of 3, 5, 7, 10, 12 and 14 days are chosen as representative to trace the forecast quality. Figure 6 displays the CRPSS values of different flood volumes at three hydrological stations. Generally, flood volumes tend to be better captured with the increase of "duration", especially for lead times from 7 day to 12 day. Performance of the VIC/ECMWF system



deteriorates with increasing lead time as expected. The effective lead time for FF is shorter than MF. This can be explained by the generation mechanism of FF. FF is usually dominated by baseflow and snowmelt. Compared with MF, FF as low-flow flood events normally occurs in the same period within one year, so historical meteorological observations on the same calendar day provide plausible inputs, and result in a reference forecast which is hard to beat. As for MF, flood can be

predicted in at least 10 days ahead. Similar to Table 2, forecasts driven by S-simulation gain higher CRPSS at NX, while for the other two stations, performance of S-simulation and N-simulations varies with lead time and "duration". It seems that N-simulations gradually lose the advantage with increasing lead time, which may have something to do with the superposition of interaction of model parameter errors and meteorological forecasts uncertainty over several parameter sets.

[Figure 6]

Another statistical indicator computed from forecasted flood volumes driven by S-simulation and N-simulations is illustrated by boxplots in Fig. 7 for FF and Fig. 8 for MF. For simplicity, only Q1 and Q7 are displayed, and the overall tendency is progressive improvement from Q1 to Q7. As can be perceived, $E_q$ increases against lead time, but for longer lead time a decrease exists. The decrease begins from day +10 for FF and day +12 for MF. Meanwhile, whiskers of boxplots become wider and wider with the increase of lead time, signaling larger degree of variability in streamflow errors over times.

Regarding comparisons between S-simulation and N-simulations, we can observe that for FF (Fig. 7) S-simulation outperforms N-simulations for forecasts verified on observations, whilst for results computed from simulations opposite performance emerges, which means that the errors from hydrological model degrade the efficacy of N-simulations. For YC and NX the hydrological model errors are dominant in the beginning and from lead time longer than 7 days errors from ECMWF forecasts prevail. However, at NGS errors from ECMWF forecasts become dominant from day +3. This difference

is probably caused by the basin scale. The hydrological response time is shorter in NGS, and the streamflow is more sensitive to meteorological inputs.

[Figure 7]

As demonstrated in Fig. 8, $E_q$ for MF is smaller than that for FF, with majority of the streamflow errors confined within 40%. Unlike FF, MF is usually dominated by the precipitation inputs during a relevant period. Accordingly, the influence from

hydrological errors becomes minor. This is why there are smaller differences between $E_q$ computed from observations and from simulations for lead time longer than 3 days. But for NGS (Fig. 12a-b), the most upstream station, the hydrological error impacts the forecasts up to day +14. Although on average performance level, forecasts derived from S-simulation certainly have smaller errors, certain cases exist where part of the members in N-simulations have the ability to provide forecasts with the smallest errors. Figure 8 also presents additional details about the ability of S-simulation and N-

simulations. For NGS and YC, at which VIC was well calibrated (Table 1), behaviors of N-simulations are inferior, while for NX, forecasts of N-simulations exhibit comparable $E_q$ with S-simulation when verifying on observations and even preferable performance verified on simulations. It seems that N-simulations scheme works in poorly-calibrated regions. Moreover,





bigger Bias and smaller NSE are present at NX (Table 1), but in term of $E_q$, the forecasts at this station are not inferior compared with the other two stations.

[Figure 8]

The errors in peak time prediction are displayed in Fig. 9. The left sides are subplots for FF, and the results for MF are

shown in right-hand subplots. Similar to $E_q$, $E_t$ deteriorates with lead time and peaks at lead time of 10 day. The peak time errors at three stations are about 1-5 days for both FF and MF, yet errors in MF are larger than that of FF. $E_t$ of FF at NGS is the largest, and the cause may be the shorter response time. The differences between the peak time errors verified on observations and simulations are not that significant compared with streamflow errors, so errors from meteorological forecasts should take most of the responsibility for $E_t$, especially for NGS and NX. In other words, the calibrated

hydrological model has a satisfactory skill in capturing the peak times. Performance of S-simulation and N-simulations in this round varies with flood categories and stations, but generally smaller errors are found in peak times forecasted by N-simulations.

[Figure 9]

### 4.3 Streamflow components forecasts

In this section, performance of the proposed HEPS in forecasting flood components is analyzed. Figures 10-12 show CRPSS of snowmelt-induced and rainfall-induced volumes at three hydrological stations. For comparison purpose, we assume that during the evaluation period the streamflow components simulated by corresponding observed meteorological inputs represent the actual components condition, and used as the observation proxy. The reference forecasts used to compute CRPSS are forecasts driven by the same parameter set with inputs of historical observations at the same calendar day. Thus,

the CRPSS here is just an indicator to show the forecast skill against lead time and to present the errors only from meteorology. Only the results for Q1 are presented, as the results show no obvious correlations with "duration".

From Fig. 10, it is noticeable that for FF at NGS, errors in forecasting surface runoff components is the main source contributing to errors in forecasting total runoff. Forecast skill for components in baseflow seems to be insensitive to lead time (Figs. 10a-b). After all, these components are mainly generated by available water storage in the catchment. As for MF,

similarly the errors derived from surface runoff forecasts are the main contributor to errors in total runoff forecasts, but the baseflow exhibits a similar tendency with surface runoff and total runoff, deteriorating with lead times as shown in Figs. 10c-d. This means during the period of MF the infiltration is substantial in VIC modeling and makes the moisture in bottom soil layer vary with the precipitation and snowmelt inputs. The information in Figs. 10c-d is in good agreement with results displayed in Fig. 10. Fluctuating CRPSS in $Q_{snow}$ and $Q_{rain}$ results in similarly fluctuating CRPSS in $Q$. The well-predicted

$Q_{rain}$ component is the critical factor for high CRPSS for total runoff. The snowmelt-induced components can be predicted with 7 days in advance for FF, and the lead time is much shorter for MF. The rainfall-induced components can be skillfully forecasted up to day +14 compared with reference forecasts.



[Figure 10]

Similar performance can be found at YC as shown in Fig. 11. Components in baseflow for FF are consistently well reproduced by the system with CRPSS greater than 0.8 for all the lead times. The variation in total runoff is fairly consistent with surface runoff. However, higher CRPSS in both $Q_{snow}$ and $Q_{rain}$ fails to give birth to higher CRPSS in $Q$ (shown in Fig.

10b). According to Table 1, the MAE value for S-simulation is 258.64 $m^3/s$ for Q1, and the average observed peaks during this period is about 630 $m^3/s$. Hence, the errors in hydrological model are too large to capture the actual flood process. The high CRPSS in this section is caused by the exclusion of hydrological errors. With regard to MF, the snowmelt-related components are forecasted with shorter lead time as well as that at NGS, and it is difficult to distinguish the system skill in different flood components.

The most noticeable phenomenon at NX is that components in baseflow for FF at this station exhibit an obvious deterioration with lead times (Fig. 12a-b). NX is located in the most downstream reaches, and concentrates water from hundreds of tributaries. In some tributaries, like Niyang River located the near upper reaches of NX station, streamflow is dominated by glacier-snow melt, perhaps leading to the baseflow and surface runoff responses rapidly, causing the baseflow in the outlet to vary with lead time. CRPSS of all the flood components has similar changes to scores of total runoff in Fig.

10. Generally, the $Q_{snow}$ and $Q_{rain}$ forecasts are skillful in lead time of 7 day and 10 day, respectively. Surface runoff remains the toughest part for forecasts, in which the snowmelt-induced components can be predicted in only 5 days ahead.

[Figure 11]  [Figure 12]

## 5. Discussion

In this study, N-Pareto-Optimal parameter sets were adopted to solve the multiple feasible solutions by multiobjective

optimization. Before NWP was introduced into the flood forecasting system, the streamflow driven by N-simulations is better simulated than that by S-simulation as shown in Table 2, although the NSE and Bias value are more favorable for S-simulation during calibration. When it comes to flood forecasting, neither of the outputs by these two simulation modes has overwhelming advantages over every aspect of forecasting, which coincides with a previous study by Zhu et al. (2016b). Three preliminary findings were made for N-simulations. Firstly, N-simulations generally behave better when the trade-off

in multi-objectives is stronger. In this case, the N-simulations can synthesize advantages from different components. This is why N-simulations provide more desirable skill at NGS than NX. Secondly, N-simulations indeed improves the streamflow simulation as shown in Table 1 and 2, but when it comes to forecasting, the interaction of errors in hydrological model parameters and meteorological forecast would degrade the forecast skill at longer lead time. Last, N-simulations may fail to provide better results on average model performance level, but individual member in N-Pareto-Optimal parameter sets can

capture the events with the lowest errors.

As mentioned before, there is no glacier module in current VIC model. The glacier-related process is considered together with the snow in this study. In other words, the rainfall input into VIC is separated into only two components, the liquid





(rainfall) and solid parts (snow), and the portion of rainfall which is supposed to turn to glacier/ice is treated as snow instead. That is why the snow depth simulated by VIC is obvious higher than that by remote sensing showed in Fig. 5 but the melt water proportion is close to the records (Table 3), for the output snow depth from VIC is actually the sum of snow and glacier/ice. Additionally, compared with the distribution of used meteorological stations in Fig. 1, we can infer that these

positive biases are also induced by the interpolation using data from stations at which there are more snow/glacier present. To verify our conclusion, we plot the VIC snow depth together with the distribution of glacier in YZR basin. The glacier data is download from The Second Glacier Inventory Dataset of China (http://westdc.westgis.ac.cn/data/f92a4346-a33f-497d-9470-2b357ccb4246). From Fig. 13, it is noticeable that the locations of overestimation do coincide with the locations of glacier. For Zone 1 and Zone 2, the overestimation is exacerbated by interpolating with gauges at which more snow and

glacier exist. To relieve this problem, there are generally two ways to considering glacier-melt separately: energy balance models to calculate melt as residual in the heat balance equation, and temperature-index models to quantify an empirical relationship between air temperature and melt rate (Zhang et al., 2013). Some studies have successfully coupled VIC with these two kinds of glacier-melt models. Zhang et al. (2013) and Su et al. (2016) embodied a simple degree-day glacier algorithm into VIC, and Zhao et al. (2013) coupled an energy balance-based glacier model with VIC, showing acceptable

performance with efficiency coefficient greater than 0.8 for the complete simulation period. However, one thing we should bear in mind is that with the limited observed data in this special area, it is difficult to accurately separate the snow from glacier. Overly complicated methods probably bring out more uncertainties. In that sense, simply dividing the input rainfall into two parts can be an acceptable way to consider meltwater as used in this study.

[Figure 13]

For streamflow components prediction, the biggest challenge is the absence of data series of streamflow components. In this way, the simulation driven by observations becomes an alternative to act as proxy, but it is difficult to determine whether such proxy is believable or not. In this way, conclusion simply based on simulation of single parameter set is risky. Similar to hydrological ensemble prediction, ensemble from multi-parameter sets is believed to be more plausible. From our results, different parameter sets behave consistently in streamflow components prediction, i.e. deterioration with increasing lead time.

However, when it comes to specific skill score, some slight differences can be viewed from Figs. 10-12. Sometimes, S-simulation provides skillful forecasts with longer lead time, while in some other cases, performance of S-simulation becomes inferior and falls out of the 95% CI. We believe that the phenomenon captured by most of the parameter sets would be the most possible truth. Single parameter set may present overestimation or underestimation to some degree.

The snow-induced components in streamflow are found to be difficult for the system to predict, in which these in surface

runoff is the toughest part. This is reasonable since the surface runoff is the most susceptible variable to various hydrometeorological factors. Specifically, $R_{snow}$ in the study area is mainly determined by the amount of snowfall and the temperature at which the snowpack begins to melt. In VIC, the inputted precipitation is separated into snowfall and rainfall according to a predefined temperature. In consequence, errors from all the ECMWF forecasts would affect the $R_{snow}$ forecasts whilst $R_{rain}$ is merely influenced by one meteorological input, QPF. This is also the reason why rainfall-induced




streamflow forecasts are the major contributor to satisfactory forecasting. This illustrates the importance of components study.

## 6. Conclusion

In this study, a hydrological ensemble prediction system composed by VIC and ECMWF medium-range precipitation and temperature forecasts was applied in YZR Basin to investigate the forecasting of flood volumes and streamflow components. Two different simulation modes were adopted. One is S-simulation which is driven by traditional single parameter set, and the other one is N-simulations which is driven by an ensemble of parameter sets selected from the Pareto front using Preference Ordering Routine method. A newly published hydrograph separation algorithm was employed to separate the streamflow into four individual components: the surface runoff and baseflow induced by rainfall and snowmelt respectively. The findings are summarized as following:

(1) N-simulations mode has been proven to be superior in model simulation. For flood forecasting, the performance of N-simulations and S-simulation varies with lead time and basin scale, and the N-simulations is recommended when the multi-objective trade-off is significant.

(2) Flood forecasts deteriorate with lead time. The forecast skill of flood volume increases with "duration". Q7 can be better captured than Q1. The forecasting system provides better forecasts for MF. The flood volume of FF can be predicted in 7-14 days in advance. The lead time for MF is 10-14 days.

(3) The meteorological error is the main source of errors in MF forecasts from lead time of +3 day, but for FF and smaller watershed (NGS) the influence of hydrological errors lasts for longer time. QPF for FF tends to be overestimated and for MF opposite bias trend is observed.

(4) Snowmelt-induced component in surface runoff is the most difficult part for the proposed system to predict compared with reference forecasts, which can only be captured in 4-7 days. Well-forecasted rainfall-induced streamflow is the main contributor for successful flood forecasting.

## Author contribution

Suli Pan provides methodology used to bias correct the raw ECMWF forecasts. Zhixu Bai helped to develop the model code. Yue-Ping Xu guided and supervised the research. Li Liu performed the simulation and prepared the manuscript with contribution from all co-authors.





**Acknowledgement**

This study is financially supported by National Natural Science Foundation of China (91547106) and National Key Research and Development Plan "Inter-governmental Cooperation in International Scientific and Technological Innovation"(2016YFE0122100). National Climate Center of China Meteorological Administration and Xizang Hydrological Bureau are greatly acknowledged for providing meteorological and hydrological data used in the study area. QPFs and temperature forecasts were obtained from ECMWF's TIGGE data portal. Thanks are also given to ECMWF for the development of this portal software and for the archives of this immense dataset.

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





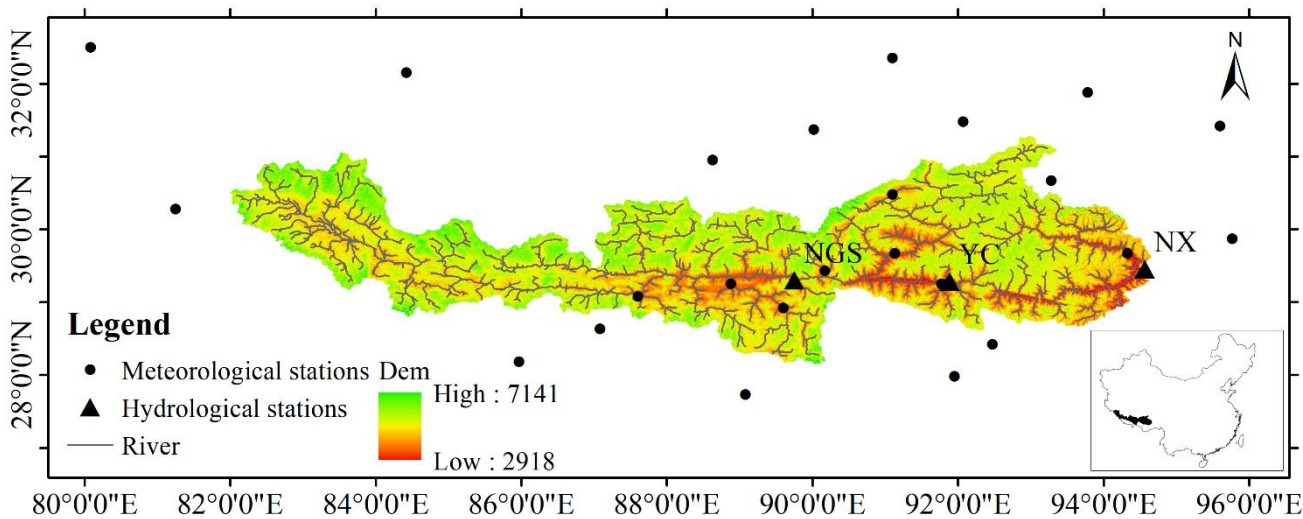

**Figure1. Location of the study area, and distribution of hydrological and meteorological stations used in this study.**

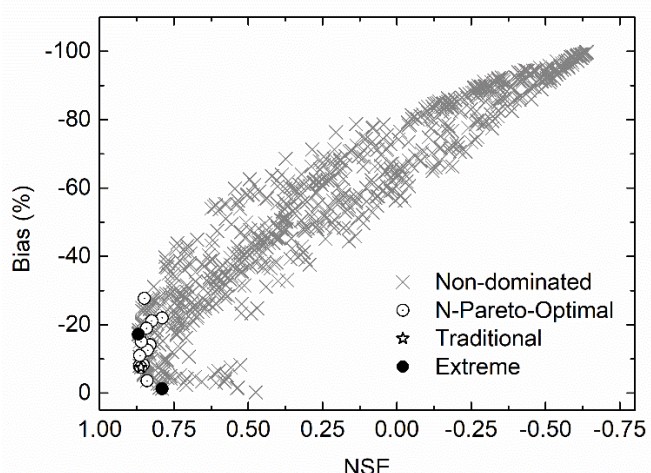

5    **Figure 2. Two-dimensional Pareto plots for Bias and NSE at NGS. The cross markers indicate all the non-dominated solutions and the circle ones are selected N-Pareto-optimal parameter sets. The traditional parameter set is denoted as star markers.**



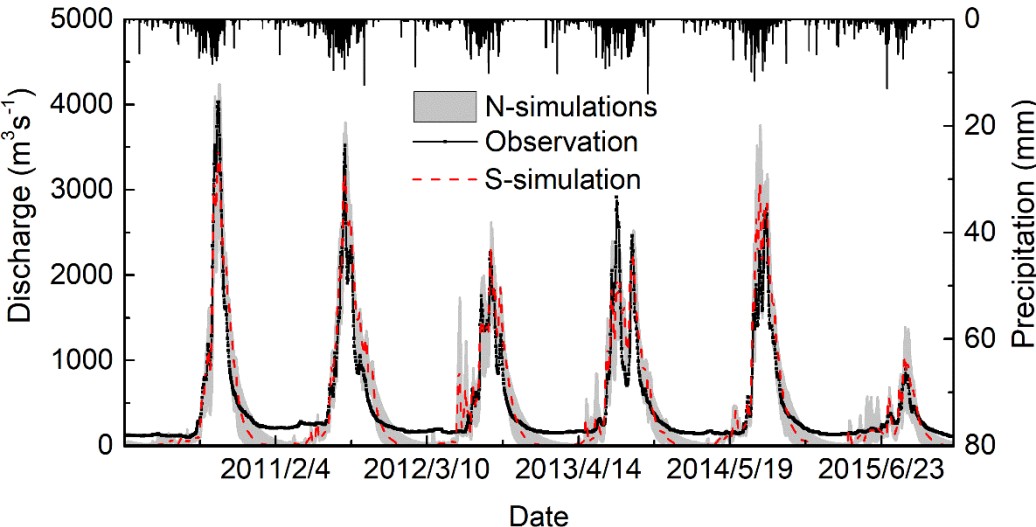

**Figure 3. Daily time series of simulated and observed streamflow at NGS. The upper bar is the areal precipitation.**

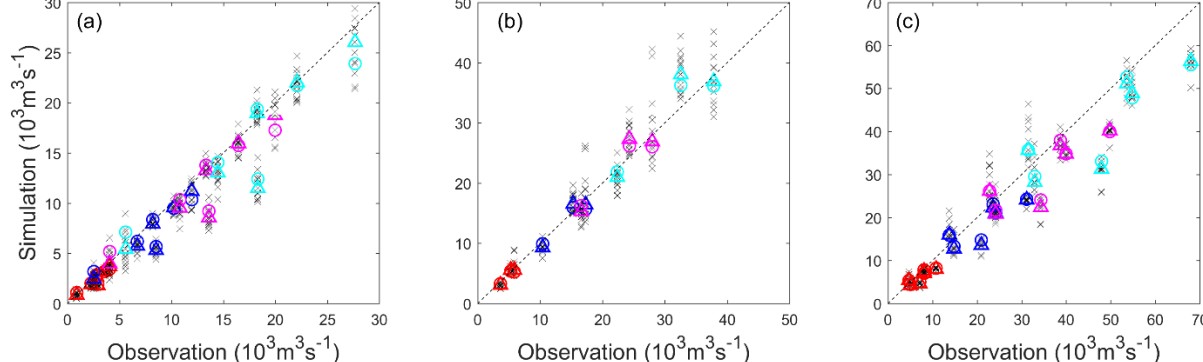

**Figure 4. Typical simulated flood volumes versus observed ones. The crosses in the figures are results by N-simulations and larger circles are results from S-simulation. The different colors are volumes for different durations, red for Q1, blue for Q3, magenta and cerulean for Q5 and Q7. (a) NGS, (b) YC, and (c) NX.**





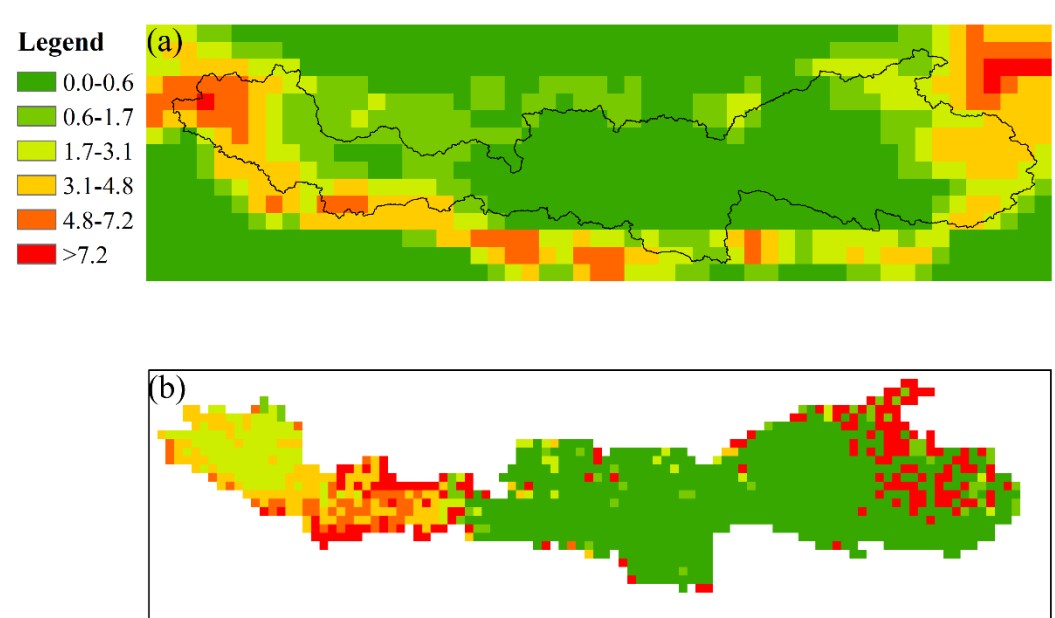

**Figure 5. Spatial distribution of daily average snow depths derived from remote-sensing (a) and simulation by S−simulation (b) at NX.**

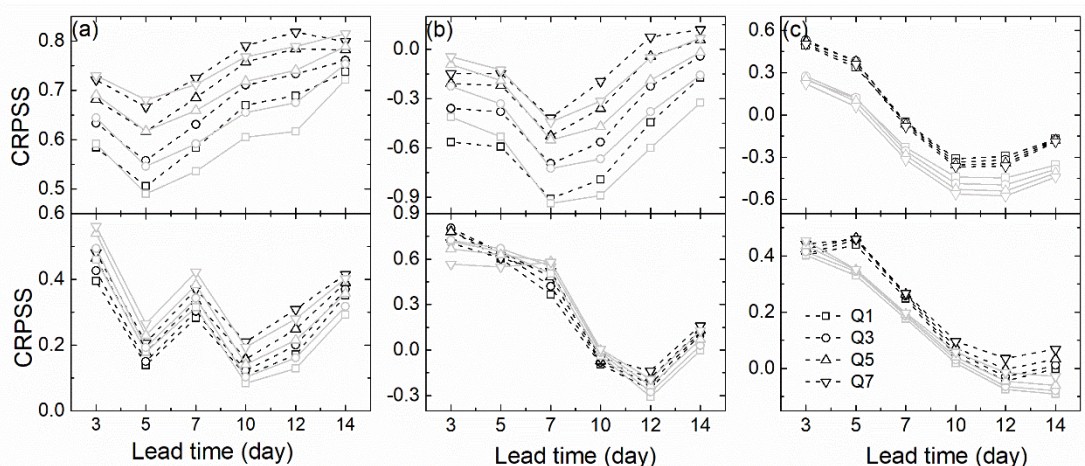

Figure 6. CRPSS for different typical accumulated flood volumes against lead times. The upper panels are results for FF and the lower ones are for MF. Scores derived from S-simulation sets are marked in black while results for N-simulations are in grey. (a) NGS, (b) YC, (c) NX.





**Figure 7.** $E_q$ **of FF for Q1 and Q7 at (a)-(b) NGS, (c)-(d) YC, and (e)-(f) NX. The first two of the four boxplots at each lead time in each subplot are results verified on observations and the remaining boxplots are verified on simulations. The unfilled boxplots are forecasts driven by S-simulation and forecasts derived from N-simulations are denoted by filled ones.**





**Figure 8.** $E_q$ **of MF for Q1 and Q7 at (a)-(b) NGS, (c)-(d) YC, and (e)-(f) NX. The first two of the four boxplots at each lead time in each subplot are results verified on observations and the remaining boxplots are verified on simulations. The unfilled boxplot are forecasts driven by S-simulation and forecasts derived from N-simulations are denoted by filled ones.**





**Figure 9.** $E_t$ **for FF and MF at (a)-(b) NGS, (c)-(d) YC, and (e)-(f) NX. The first two of the four boxplots at each lead time in each subplot are results verified on observations and the remaining boxplots are verified on simulations. The unfilled boxplot are forecasts driven by S-simulation and forecasts derived from N-simulations are denoted by filled ones.**



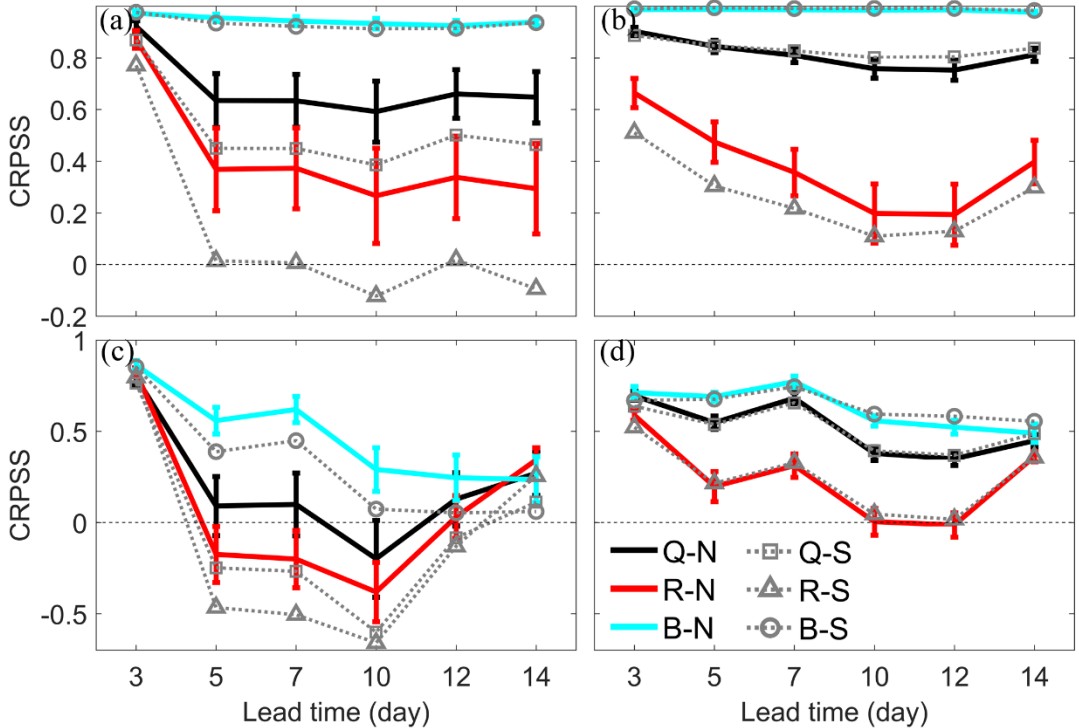

**Figure 10. CRPSS of four different streamflow components against lead time at NGS. Snowmelt-induced components for FF (a) and MF (c), rainfall-induced components in FF (b) and MF (d). The thick and solid lines are CRPSS by N-simulations with vertical bars showing 95% confidence intervals and the dashed lines with different markers are CRPSS by S-simulation. Black lines are snowmelt/rainfall components in total runoff (Q). Red lines are CRPSS for components in surface runoff (R) and blue ones are in base flow (B).**





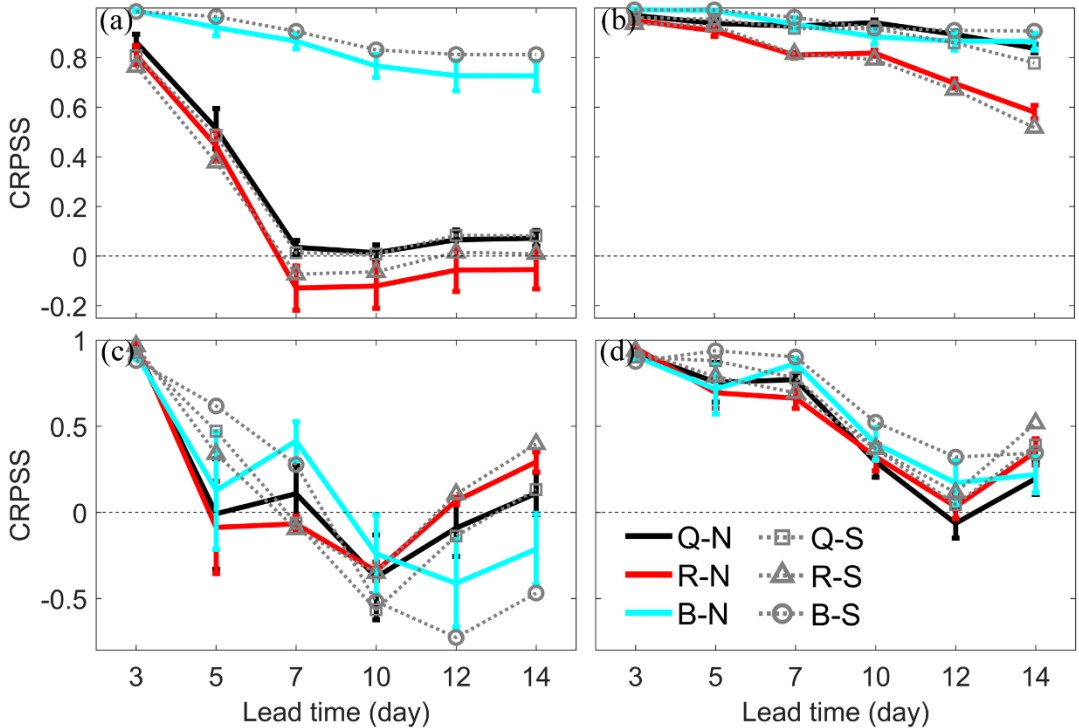

**Figure 11. CRPSS of four different streamflow components against lead time at YC. Snowmelt-induced components for FF (a) and MF (c), rainfall-induced components in FF (b) and MF (d). The thick and solid lines are CRPSS by N-simulations with vertical bars showing 95% confidence interval and the dashed lines with different markers are CRPSS by S-simulation. Black lines are snowmelt/rainfall components in total runoff (Q). Red lines are CRPSS for components in surface runoff (R) and blue ones are in base flow (B).**




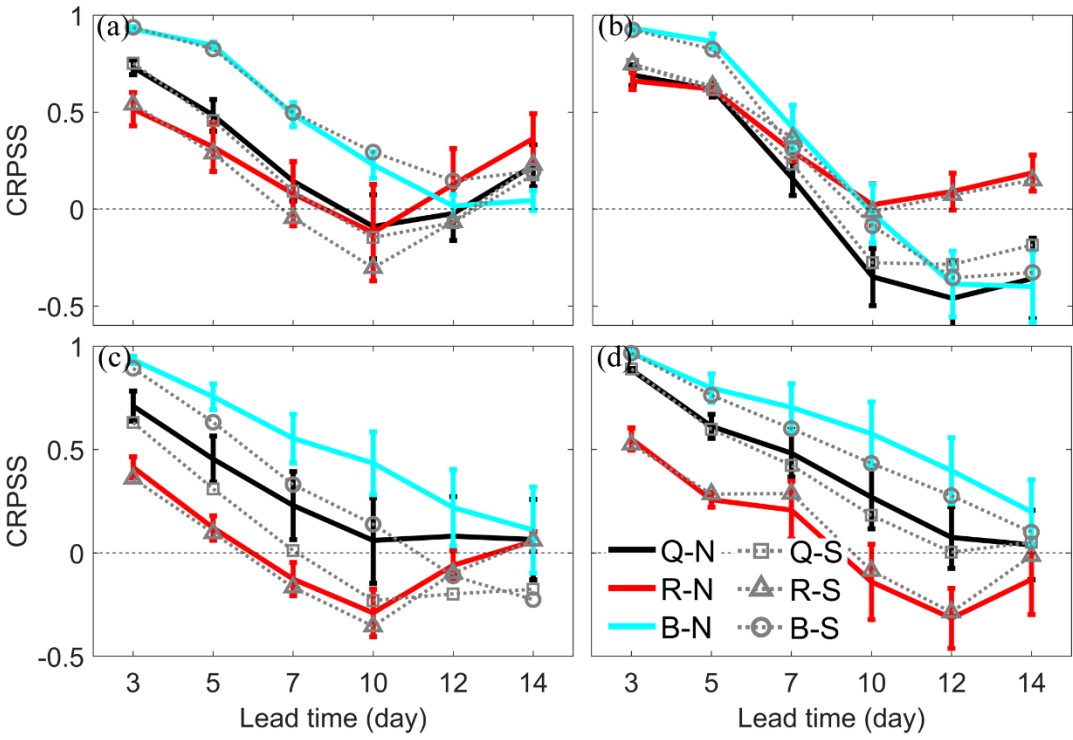

**Figure 12. CRPSS of four different streamflow components against lead time at NX. Snowmelt-induced components for FF (a) and MF (c), rainfall-induced components in FF (b) and MF (d). The thick and solid lines are CRPSS by N-simulations with vertical bars showing 95% confidence intervals and the dashed lines with different markers are CRPSS by S-simulation. Black lines are snowmelt/rainfall components in total runoff (Q). Red lines are CRPSS for components in surface runoff (R) and blue ones are in base flow (B).**

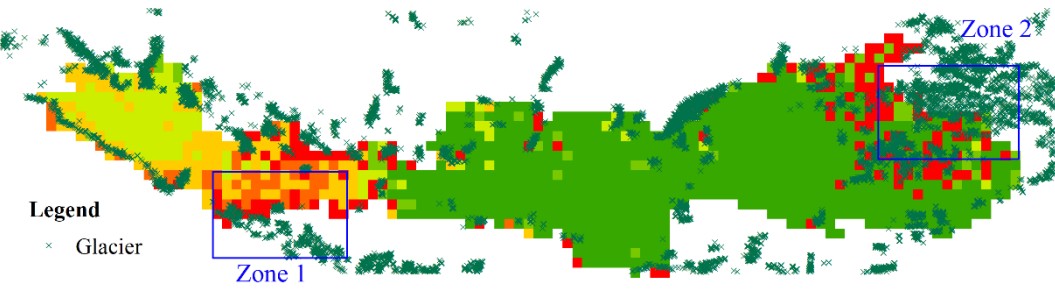

**Figure 13. Spatial distribution of VIC snow depth and glacier in YZR basin.**



**Table 1. Information of N-simulations and S-simulation During Calibration and Evaluation at Three Hydrological Stations.**

| Station | Numbers | Mode | Calibration/Evaluation | | | |
|---------|---------|------|------|---------|--------|----------|
| | | | NSE | Bias(%) | NSE10% | Bias10%(%) |
| NGS | 16 | N-simulations | 0.77~0.87/ 0.77~0.88 | -27.75~-1.30/ -21.72~6.37 | 0.06~0.51/ -0.05~0.48 | -8.83~8.05/ -9.29~16.38 |
| | | S-simulation | 0.86/0.86 | -7.55/-2.74 | **0.51/0.48** | -3.02/1.29 |
| YC | 15 | N-simulations | 0.71~0.88/ -0.07~0.65 | -34.03~-10.52/ -17.72~6.37 | -1.11~0.34/ -1.41~0.73 | -14.21~2.60/ -9.29~16.38 |
| | | S-simulation | **0.88**/0.56 | -13.54/-8.81 | 0.32/**0.73** | -7.43/-9.29 |
| NX | 11 | N-simulations | 0.65~0.77/ 0.58~0.79 | -44.33~-34.82/ -46.53~-34.45 | -1.27~-0.45/ -0.87~0.23 | -27.83~-20.06/ -16.36~-4.17 |
| | | S-simulation | **0.77**/0.74 | -35.03/-35.51 | **-0.45**/0.06 | **-20.06**/-5.33 |



**Table 2. CRPS and MAE for N-simulations and S-simulation on four typical flood volumes indexes during the whole period. The results are displayed as MAE/CRPS. CRPS is the indicator used for N-simulations. MAE is used for S-simulation.**

| Events | Volumes | MAE/CRPS | | |
|--------|---------|----------|-----|-----|
|        |         | NGS | YC | NX |
| FF | Q1 | 107.65/96.42 | 258.64/230.82 | 315.74/379.21 |
|    | Q3 | 297.30/266.81 | 714.26/636.85 | 795.62/998.83 |
|    | Q5 | 461.82/409.22 | 1089.45/976.46 | 1181.44/1517.56 |
|    | Q7 | 611.13/530.84 | 1412.74/1274.65 | 1524.84/2010.17 |
| MF | Q1 | 537.88/467.14 | 818.24/731.23 | 1824.27/2025.75 |
|    | Q3 | 1497.96/1267.92 | 2280.90/2021.00 | 5125.15/5608.94 |
|    | Q5 | 2304.14/1919.31 | 3471.46/3081.09 | 7820.15/8514.79 |
|    | Q7 | 3016.17/2514.06 | 4438.17/3975.66 | 10091.79/10940.98 |



**Table 3. Fractions of snowmelt-induced streamflow to total runoff during the evaluation period for three stations.**

| Station | Recorded | Simulated | |
|---------|----------|--------------|--------------|
|         |          | N-simulation | S-simulation |
| NGS     | 18%      | 14%~25%      | 16%          |
| YC      | 20%      | 11%~30%      | 25%          |
| NX      | 38%      | 20%~37%      | 35%          |