# Peer review of "Potential application of hydrological ensemble prediction in forecasting flood and its components over the Yarlung Zangbo River Basin, China"

_Hydrology and Earth System Sciences, 2018_

## Referee Comment (RC1) · R. Romanowicz (Referee) · 9 May 2018

The paper presents an application of ensemble meteorological forecasts from the ECMWF to forecasting flood volumes and other streamflow components in the YZR Basin.

The application of flow separation in forecasting and analysis of forecast skills of different flow components, including flood volumes, base flow, first flood in a year and annual maximum flood are the main novelty of the study. The second aim of the paper is to study "the impact of an ensemble of Pareto optimal solutions on model simulations". I feel that the authors have failed in combining those aims together and it is the

main drawback of the paper.

The authors apply the Variable Infiltration Capacity model (VIC) to ensemble flood forecasting and to separate snow-driven runoff from the total runoff. The interpolation of daily data to 6-h time step was performed using a Inverse Distance Weighted method coupled with an elevation-based lapse rate.

The available observations were divided into calibration, verification and evaluation periods. The authors refer to their previous paper published in Journal of Hydrology (Liu et al., 2017) when listing the additional datasets used in the present paper. I advise them to repeat that information in the present paper to help the reader. Also the authors refer to that paper while describing the calibration stage and as a result the description has become not very transparent. It is not clear if the snowmelt component was previously used and the number and names of parameters optimised in the present paper are missing. The authors should state clearly which parameters they optimise and how the separation into snow-induced runoff component is calibrated. The authors mention some validation but the description is not clear. In summary, the Data section should be extended and the Methodology section re-organised.

The authors compare an ensemble of multiple objective Pareto simulations and single best in their ability to forecast different flood components. The authors apply Preference Ordering Routine (POR) to choose the N-Pareto-optimal sets. There is no explanation of why that particular method was used nor what is the physical meaning of the applied numerical procedure. The authors set the number of ensemble members to ten, but it is not explained why that number was chosen in statistical terms.

In the hydrograph separation subsection 3.2, the authors do not explain which data were used for the calibration/validation of the separation parameters.

The post-processing of ECMWF forecasts is performed in an arbitrary way, without checking if it is necessary and provides better forecasts. Bias correction does not always give positive results regarding forecasting (Kiczko et al., 2015, Benninga et al.,

2018).

The Results section includes hydrological model performance and an assessment of flood volume and flood component forecasts. This section I find very confusing. It does not help that the authors use abbreviations that the reader needs to be acquainted with.

The authors conclude that 7-day accumulated flood volumes are easier to forecast than the peak flows. The snow-induced flood component is not well captured whilst the rainfall-induced floods are forecast well. Taking into account the fact that the snow and glacier melt forecasts were not available that conclusion is not surprising.

The authors find that the base flow component forecast is insensitive to the forecast lead. As the base flow dynamics is slower, the forecast lead may not cover the base flow variability. However, on page 12, the authors state that for NX, the base flow forecasts show a deterioration with a lead time. A synthesis of the overall results is missing.

The language requires correcting by a native English speaker. The text is rather difficult to follow.

Overall I find the paper interesting and worth publishing after moderate corrections, despite the need to improve the presentation of the text and the language problems.

Some Editorial comments:

Page 1: The Abstract conclusions are not transparent. It is not mentioned that the forecasting performance varies within the catchments studied.

Page 2, lines 5-9 Style should be corrected.

Page 2., lines 25-27: Is this snow tracking model used in the present paper?

Page 3, lines 111-13: style should be corrected.
Page 5, lines 17 and 20: instead of the word "theorem" I would use "attribute"

Pages 7- to the end: there are language problems in nearly all pages and language editing by a native English speaker is needed.

Figure 5 – there should be some quantitative assessment of the differences between simulated and observed snow cover. The comparison does not look well!

Figures 10-12 - it would help if the columns were named (snow-melt -induced and rainfall-induced components).

Benninga et al. (2018) is not referred to in the text.

References

Benninga, H-J., Booij, M.J., Romanowicz, R.J., Rientjes T.H.M, (2017) Performance of ensemble streamflow forecasts under varied hydrometeorological conditions, Hydrol. Earth Syst. Sci., 21, 5273–5291, https://doi.org/10.5194/hess-21-5273-2017

Kiczko, A., Romanowicz, R.J., Osuch, M., Pappenberger, F., 2015, Adaptation of the Integrated Catchment System to On-line Assimilation of ECMWF Forecasts, in: R.J. Romanowicz and M. Osuch (eds.), Stochastic Flood Forecasting System, GeoPlanet: Earth and Planetary Sciences, Springer International Publishing, DOI 10.1007/978-3-319-18854-6_1, 173-186

---

## Referee Comment (RC2) · Y. Tian (Referee) · 5 Jun 2018

The manuscript provides a study investigating the ensemble flood forecasting based on VIC macroscale hydrological model and ECMWF numerical weather prediction in a large snow-present basin located in Tibetan Plateau. The interesting part is that the authors involve an evaluation of impact of parameter uncertainty on streamflow components, to be exactly, the snow-induced and rainfall-induced components. Overall, the objective of the paper is clear and the results are interesting. However, several issues are unclear and/ or not sufficiently considered/ described. I firmly believe it deserves to be published in the Hydrology and Earth System Sciences

subject to several moderate revisions. Please find my main comments and detailed comments below. 1. The main issue in this manuscript is that the authors need more efforts to outline the advantages and disadvantages by using multiple parameter sets (N-simulations in the manuscript). It seems that more attentions are focused on the influence of multiple parameter sets on entire streamflow. I would suggest the authors added more contents in discussion to clarify the merits of N-simulation for streamflow components forecasting and evaluating. 2. The main part in current manuscript is the evaluation of N-simulations for entire streamflow. In my opinion, the evaluation of entire streamflow is the foundation of evaluation of streamflow components. For there is no direct reference data for evaluation of streamflow components. The accuracy in entire streamflow is to some degree the evidence of model ability. This is the reason for usage of simulated streamflow components driven by observation as reference in components evaluation. I suggest the authors clarify this in the manuscript. 3. A brief data description used for hydrological modelling should be supplemented though it is similar for the previous publication. 4. I would suggest the authors to simplify the contents about hydrograph separation. Detailed description can be found in Li et al. (2017), and it is better for the authors to just list the differences for the method used in this manuscript with the original one. 5. There is no need to verify the forecasts with two reference data. Evaluation based on simulations is irrelevant to the objectives mentioned in introduction. I suggest omitting the related parts. If it remains, I would like to see it more strongly justified 6. I could understand most of the manuscript without difficulty, and the methods are well documented. However, there are many minor errors in spelling, grammar and English style that I have not corrected. I recommend that the authors have the manuscript proof-read by a native English speaker before publication. Specific comments: 7. Page 6, Line 4: as snow and glacier melt water is considered together, term "meltwater" is better representative than "snowmelt". 8. Page 9, Line 31: "Lead times of 3, 5, 7, 10, 12 and 14 days" make it "Lead times of day 3, 5, 7, 10, 12 and 14" 9. Page 9, Line 32-33: "Generally, flood volumes tend to be better captured with the increase of "duration", especially for lead times from 7 day

to 12 day" should be explained. 10. Page 10, Line 32: "It seems that N-simulations scheme works in poorly-calibrated regions." from what results the authors draw this conclusion. 11. Page 13, Lines 2: "but" make it "while" 12. Page 13, Lines 27: "We believe that the phenomenon captured by most of the parameter sets would be the most possible truth." Change it to "From the view of ensemble, the phenomenon captured by more parameter sets is regarded as the most possible occurrence."

Please also note the supplement to this comment:
https://www.hydrol-earth-syst-sci-discuss.net/hess-2018-179/hess-2018-179-RC2-supplement.pdf

---

## Author Comment (AC1) · 13 Jul 2018

Dear Editor and referees,

Thanks a lot for your great efforts to read through this manuscript and give very valuable comments. We agree with your suggestions which will be of great help to improve the quality of our manuscript. Here we have addressed the comments from you and the detailed description is attached in this document.

Best regards, Li Liu, Suli Pan, Zhixu Bai, Yue-Ping Xu

Response to main comments:

[Figure]

1. The application of flow separation in forecasting and analysis of forecast skills of different flow components, including flood volumes, base flow, first flood in a year and annual maximum flood are the main novelty of the study. The second aim of the paper is to study "the impact of an ensemble of Pareto optimal solutions on model simulations". I feel that the authors have failed in combining those aims together and it is the main drawback of the paper. Response: Thank you very much for your comments. We agree that the original manuscript fails to combine the two aims properly. As the streamflow components are unknown, a plausible total runoff doesn't mean accurate streamflow components. Under this circumstance, we attempt to capture the most possible flow components by applying an ensemble of parameters to take account of more scenarios. The accuracy of multiple parameters in total runoff is the precondition for application in further flow components forecasts. Thus, the evaluation of N-simulations (simulation from ensemble of multiple parameters) in Subsection 4.1 and 4.2 is to demonstrate the efficacy of N-simulations and to prove the feasibility for flow components simulation. However, we didn't mention it in the original manuscript and make the paper somewhat confused and fragmented. Our solution is to clarify the purpose to adopt an ensemble of Pareto optimal parameter for flow components forecasting to make the paper more logical and integral.

2. The available observations were divided into calibration, verification and evaluation periods. The authors refer to their previous paper published in Journal of Hydrology (Liu et al., 2017) when listing the additional datasets used in the present paper. I advise them to repeat that information in the present paper to help the reader. Also, the authors refer to that paper while describing the calibration stage and as a result the description has become not very transparent. It is not clear if the snowmelt component was previously used and the number and names of parameters optimised in the present paper are missing. The authors should state clearly which parameters they optimize and how the separation into snow-induced runoff component is calibrated. The authors mention some validation but the description is not clear. In summary, the Data section should be extended and the Methodology section re-organised. Response:

Thanks for your comments. We are so sorry to omit the relevant data description and methodology introduction in the original manuscript and we agree that this part should be rephrased and reorganized. Different from the previously published paper, in current study the snow model related parameters are calibrated with the other normally calibrated parameter and detailed calibrated parameter will be clarified in Subsection 3.1 in the revised manuscript. The separation into snow-induced runoff components is calibrated based on two model parameters: the maximum temperature at which snow can fall and the minimum temperature at which rain can fall. These two parameters can separate the input precipitation into two parts: the liquid and the iced portion, and furtherly the total runoff is separated following the hydrological separation in Subsection 3.2.

3. The authors compare an ensemble of multiple objective Pareto simulations and single best in their ability to forecast different flood components. The authors apply Preference Ordering Routine (POR) to choose the N-Pareto-optimal sets. There is no explanation of why that particular method was used nor what is the physical meaning of the applied numerical procedure. The authors set the number of ensemble members to ten, but it is not explained why that number was chosen in statistical terms. Response: The explanation for why the POR method was adopted and why the number of ten was set will be added in the revised manuscript. In practical sense, users of automatic calibration routines have to face the task of selecting a set of suitable model parameters (preferred solution set) from the numerous Pareto-optimal sets. This is also the challenge for our study. The POR is proposed exactly to solve this kind problem by sorting out small number of preferred solutions. This method has been demonstrated for calibration of MIKE11/NAM rainfall-runoff model and is able to provide the best estimated parameter sets with good overall model performance (Khu, 2005). These are the reasons why POR was adopted in this study. The number of 10 is set mainly for the consideration of computational capacity. The number of parameter sets would be more than 10 when superadded the extreme value and the compromise value in any two-objective trade-off. Given the large number of ensemble meteorological forecasts

Interactive
comment

and different lead times, the parameter sets less than 20 would be manageable and achievable. In the work of Khu (2005), two solutions with efficiency of order 3 with degree 4 are able to provide still good performance, so in this study, the number of 10 is enough for representative of model parameter. As a matter of fact, when applying POR for all possible combinations of the four objective functions in this study, the final number of preferred parameter set is less than 10. Though we presupposed the upper limit, the final results didn't reach it.

4. In the hydrograph separation subsection 3.2, the authors do not explain which data were used for the calibration/validation of the separation parameters. Response: There is no parameters needed to be calibrated in Subsection 3.2. All the variables in the formulas are either known (model outputs) or unknown but able to be obtained by an iteration process like $f\_(W,snow,t)$. The only related parameters are the Maximum Temperature at which snow can fall and the Minimum Temperature at which rain can fall. These two parameters determine the amount of rainfall ãĂŰRainãĂŮ\_t and snowmelt $M\_t$ in Eq. (5) ($R\_(snow,t)= R\_t f\_(R,snow,t)= R\_t f\_(i,snow,t)= R\_t M\_t/(M\_t+$ãĂŰRainãĂŮ\_t)) and are calibrated with other soil parameters in VIC based on streamflow related objective functions (Eq. (1)-(4)).

5. The post-processing of ECMWF forecasts is performed in an arbitrary way, without checking if it is necessary and provides better forecasts. Bias correction does not always give positive results regarding forecasting (Kiczko et al., 2015, Benninga et al., 2018). Response: As a matter of fact, we conducted a preliminary analysis of the performance of parameterized QM on ECMWF forecasts and we found that the CRPS from bias-corrected forecasts is much smaller than that from the raw forecasts especially for temperature forecasts. Given the already redundant figures in the original manuscript, we left them out in the submission. If the referee thinks it is necessary to include the evaluation of NWP forecast in the paper, we would like to add them in the revised manuscript or just add some comments about the post-processing for simplicity.

6. The Results section includes hydrological model performance and an assessment of

flood volume and flood component forecasts. This section I find very confusing. It does not help that the authors use abbreviations that the reader needs to be acquainted with. Response: Thanks for your comments. Originally, we thought the abbreviations would be beneficial for the readers to read and understand the manuscript. However, according to the suggestions from the referee, we will reduce the abbreviations in the revised manuscript. The hydrological station names and the flood events can be indicated by full name which would be helpful for reading and make less abbreviations that the reader needs to be acquainted with. At the same time, we will improve our expression and wording to make the paper more readable.

7. The authors conclude that 7-day accumulated flood volumes are easier to forecast than the peak flows. The snow-induced flood component is not well captured whilst the rainfall-induced floods are forecast well. Taking into account the fact that the snow and glacier melt forecasts were not available that conclusion is not surprising. Response: We agree with you that since snow and glacier melt forecasts are not available, the performance in these streamflow components is unsurprisingly inferior due to various uncertainties. We know that the snow/glacier melting is influenced by not only the input rainfall and temperature but also the ability of model to capture melting process. However, considering the rainfall-induced components are also unavailable and the portion of rainfall used to generate this component is determined by the same procedure used to determine the snowfall to generate snowmelting, the comparison between these components can tell some relatively valid conclusion. We will add the reason you suggested in the revised manuscript and it would make the conclusion more strongly justified.

8. The authors find that the base flow component forecast is insensitive to the forecast lead. As the base flow dynamics is slower, the forecast lead may not cover the base flow variability. However, on page 12, the authors state that for NX, the base flow forecasts show a deterioration with a lead time. A synthesis of the overall results is missing. Response: We totally agree with the referee that the insensitivity of base

flow to lead time is also caused by the slower flow dynamics and the lead time doesn't cover the flow variability. We will add this possible reason in the revised manuscript to make the study strongly justified. Thanks for the reminding. Due to our negligence, the unique behavior of base flow at NX is missing in the conclusion. We will add the contents related to NX into the final section and make the conclusion more accurate and thorough.

9. The language requires correcting by a native English speaker. The text is rather difficult to follow. Response: We are so sorry and the manuscript will be carefully checked and polished by the native English speaker.

Response to Editorial comments:

1. Page 1: The Abstract conclusions are not transparent. It is not mentioned that the forecasting performance varies within the catchments studied. Response: The forecasting performance does vary for different sub-catchments and the detailed difference in three sub-area will be added in the Abstract. The main difference for three sub-catchments is that baseflow components at NX tends to change with the lead time.

2. Page 2, lines 5-9 Style should be corrected. Response: We will correct the style for lines 5-9 in Page 2.

3. Page 2., lines 25-27: Is this snow tracking model used in the present paper? Response: Yes, this snow tracking model is used in this study.

4. Page 3, lines 111-13: style should be corrected. Response: We will correct the style for lines 111-113 in Page 3.

5. Page 5, lines 17 and 20: instead of the word "theorem" I would use "attribute". Response: We will replace the word "theorem" with "attribute" in the revised manuscript.

6. Pages 7- to the end: there are language problems in nearly all pages and language editing by a native English speaker is needed. Response: Native English speaker will be asked to check and polish the entire manuscript.

7. Figure 5 – there should be some quantitative assessment of the differences between simulated and observed snow cover. The comparison does not look well! Response: We will add the quantitative assessment for the simulated and observed snow cover in the revised manuscript. The special correlation coefficient and the overall bias in entire study area will be calculated to show more direct evaluation.

8. Figures 10-12 - it would help if the columns were named (snow-melt -induced and rainfall-induced components). Response: We will name the columns in Figures 10-12 to make the figure more accessible.

9. Benninga et al. (2018) is not referred to in the text. Response: The reference for Benninga et al. (2018) will be added in the revised manuscript.

Please also note the supplement to this comment:
https://www.hydrol-earth-syst-sci-discuss.net/hess-2018-179/hess-2018-179-AC1-supplement.pdf

---

## Author Comment (AC2) · 13 Jul 2018

Dear Editor and referees,

Thanks a lot for your great efforts to read through this manuscript and give very valuable comments. We agree with your suggestions which will be of great help to improve the quality of our manuscript. Here we have addressed the comments from you and the detailed description is attached in this document.

Best regards, Li Liu, Suli Pan, Zhixu Bai, Yue-Ping Xu

Response to main comments:

[Figure]

1. The main issue in this manuscript is that the authors need more efforts to outline the advantages and disadvantages by using multiple parameter sets (N-simulations in the manuscript). It seems that more attentions are focused on the influence of multiple parameter sets on entire streamflow. I would suggest the authors added more contents in discussion to clarify the merits of N-simulation for streamflow components forecasting and evaluating.

Response: We agree that more attention should be paid to clarify the advantages and disadvantages for total runoff and component flow by using multiple parameter sets. In the original manuscript, we seem to give excessive description for entire streamflow assessment. Actually, the evaluation of multiple parameter sets on entire streamflow is to verify the feasibility and applicability of N-simulations for streamflow components. As the observed component flows are unavailable, the ability in simulating entire flow to some degree indicates the ability for component flows. According to the suggestion from the referee, we will focus more on Figures 10-12 to illustrate the merits of N-simulations for component flow forecasts.

2. The main part in current manuscript is the evaluation of N-simulations for entire streamflow. In my opinion, the evaluation of entire streamflow is the foundation of evaluation of streamflow components. For there is no direct reference data for evaluation of streamflow components. The accuracy in entire streamflow is to some degree the evidence of model ability. This is the reason for usage of simulated streamflow components driven by observation as reference in components evaluation. I suggest the authors clarify this in the manuscript.

Response: Thanks for this very useful suggestion. The evaluation of N-simulations for entire streamflow is exactly the base of evaluation for component flows, which is not mentioned in the original manuscript. To make the paper more logical, we will add this interpretation in the subsection 4.1, the first beginning for results. We believe that this will make the paper easier to understand.

3. A brief data description used for hydrological modelling should be supplemented though it is similar for the previous publication.

Response: As mentioned in response to Referee #1, snow model and frozen algorithm are used in current study, which is different to previous study. In this way, the description of related snow depth data and additional calibration parameters will be added in Subsection 2.2 and 3.1 respectively.

4. I would suggest the authors to simplify the contents about hydrograph separation. Detailed description can be found in Li et al. (2017), and it is better for the authors to just list the differences for the method used in this manuscript with the original one.

Response: We will simplify the introduction of hydrological separation, but the subsection will be retained for this is a key component for our study. In the research of Li et al. (2017), only the snow induced components in total runoff is calculated, while in this study the streamflow is separated into four different parts. This difference will be depicted in detail in the revised manuscript.

5. There is no need to verify the forecasts with two reference data. Evaluation based on simulations is irrelevant to the objectives mentioned in introduction. I suggest omitting the related parts. If it remains, I would like to see it more strongly justified

Response: We agree that the verification based on two reference data is to some degree irrelevant to the two aims mentioned in introduction. We will revise the manuscript by using only the observed streamflow as verification data which is supposed to make the paper clearer.

6. I could understand most of the manuscript without difficulty, and the methods are well documented. However, there are many minor errors in spelling, grammar and English style that I have not corrected. I recommend that the authors have the manuscript proof-read by a native English speaker before publication.

Response: The manuscript will be carefully checked and polished by a native English

speaker.

Response to specific comments:

7. Page 6, Line 4: as snow and glacier melt water is considered together, term "melt-water" is better representative than "snowmelt".

Response: Thanks for your suggestion. The term "snowmelt" will be replaced by "melt-water" in the revised manuscript.

8. Page 9, Line 31: "Lead times of 3, 5, 7, 10, 12 and 14 days" make it "Lead times of day 3, 5, 7, 10, 12 and 14"

Response: "Lead times of 3,5,7,10,14 days" will be changed into "Lead times of day 3, 5,7,10,12,14".

9. Page 9, Line 32-33: "Generally, flood volumes tend to be better captured with the increase of "duration", especially for lead times from 7 day to 12 day" should be explained.

Response: The detailed explanation will be added. There are two reasons for why flood volumes tend to be better captured with the increase of "duration". One is that there are errors in peak time which makes the single day flood volume prone to bias. Another reason is that when the "duration" increases, the bias from streamflow for this relatively long period can offset with each other.

10. Page 10, Line 32: "It seems that N-simulations scheme works in poorly-calibrated regions." from what results the authors draw this conclusion.

Response: When we compared Figures 7-9, we found that NX is the only case that N-simulations consistently provide comparable or even better results than S-simulation when verified on simulated streamflow. Since the contents related to evaluation on simulated streamflow will be left out due to the irrelevance to study topic as suggested by referee, this sentence will be deleted in the revised manuscript.

11. Page 13, Lines 2: "but" make it "while"

Response: We will change "but" into "while".

12. Page 13, Lines 27: "We believe that the phenomenon captured by most of the parameter sets would be the most possible truth." Change it to "From the view of ensemble, the phenomenon captured by more parameter sets is regarded as the most possible occurrence."

Response: Thanks for your suggestion. The sentence will be revised into "From the view of ensemble, the phenomenon captured by more parameter sets is regarded as the most possible occurrence."

Please also note the supplement to this comment:
https://www.hydrol-earth-syst-sci-discuss.net/hess-2018-179/hess-2018-179-AC2-supplement.pdf

―――――――――――――――――――

---

## Author Response (AR1)

**Revision notes for "Potential application of hydrological ensemble prediction in forecasting flood and its components over the Yarlung Zangbo River Basin, China" (hess-2018-179)**

Dear Editor and referees,

Thanks a lot for your great efforts to read through this manuscript and give very valuable comments. We agree with your suggestions which will be of great help to improve the quality of our manuscript. Here we have addressed the comments from you and the detailed description is attached in this document.

10 Best regards,
Li Liu, Yue-Ping Xu, Suli Pan, Zhixu Bai

**To Referee 1**

**Response to main comments:**

15 **1**. The application of flow separation in forecasting and analysis of forecast skills of different flow components, including flood volumes, base flow, first flood in a year and annual maximum flood are the main novelty of the study. The second aim of the paper is to study "the impact of an ensemble of Pareto optimal solutions on model simulations". I feel that the authors have failed in combining those aims together and it is the main drawback of the paper.

**Response:** Thank you very much for your comments. We agree that the original manuscript fails to combine the two aims
20 properly. As the streamflow components are unknown, a plausible total runoff doesn't mean accurate streamflow components. Under this circumstance, we attempt to capture the most possible flow components by applying an ensemble of parameters to take account of more scenarios. The accuracy of multiple parameters in total runoff is the precondition for application in further flow components forecasts. Thus, the evaluation of N-simulations (simulation from ensemble of multiple parameters) in Subsection 4.1 and 4.2 is to demonstrate the efficacy of N-simulations and to prove the feasibility for flow components
25 simulation. However, we didn't mention it in the original manuscript and make the paper confused and fragmented. Our solution is to clarify the purpose to adopt an ensemble of Pareto optimal parameter for flow components forecasting to make the paper more logical and integral. Thus, we added some clarifications in our revised paper.

Page 2, Lines 28-32: "Generally, evaluating model performance should be performed based on in-situ observations. However, observed streamflow components are usually unavailable, making the evaluation of streamflow component
30 simulations/forecasts intractable. The alternative solution is to verify forecasts on model simulations assuming that simulations

driven by meteorological observations present the actual hydrological components. However, the success of this practice highly depends on how well the hydrological model is calibrated…"

Page 7, Lines 3-6: "As a result of unavailability of observed streamflow components, the evaluation of streamflow components has to be done based on the facts that the total runoff is accurately simulated and forecasted, and that the ratio of meltwater-

5 induced streamflow is similar to records in previous studies. Hence, a general assessment on total runoff simulations and forecasts is crucial and thus done first in Subsection 4.1 and 4.2…"

Page 12, Lines 5-11: "…From analysis above, an encouragingly accurate VIC simulation and forecasting system is established in YZR. This is an important precondition for subsequent evaluation of streamflow components. In some cases, for example first floods at Yangcun, VIC fails to produce accurate enough simulations and thus poor forecasts (Fig. 6b and Fig. 7c). When

10 discussing streamflow components in these circumstances, we should evaluate model performance more carefully with consideration of errors possibly stemming from hydrological model, although the comparison of streamflow component forecasts is performed based on simulated components and to some degree, the errors in forecasts are mainly subject to meteorological inputs and therefore the hydrological error becomes negligible…"

15 **2.** The available observations were divided into calibration, verification and evaluation periods. The authors refer to their previous paper published in Journal of Hydrology (Liu et al., 2017) when listing the additional datasets used in the present paper. I advise them to repeat that information in the present paper to help the reader. Also, the authors refer to that paper while describing the calibration stage and as a result the description has become not very transparent.        It        is        not        clear        if        the        snowmelt        component

20 was previously used and the number and names of parameters optimised in the present paper are missing. The authors should state clearly which parameters they optimize and how the separation into snow-induced runoff component is calibrated. The authors mention some validation but the description is not clear. In summary, the Data section should be extended and the Methodology section re-organised.

**Response:** We are so sorry to omit the relevant data description and methodology introduction in the original manuscript and

25 we agree that this part should be rephrased and reorganized. Different with the previously published paper, in current study the snow model related parameters are calibrated with the other normally calibrated parameter and detailed calibrated parameter will be clarified in Subsection 3.1 in the revised manuscript. The separation into snow-induced runoff components is calibrated based on two model parameters: the maximum temperature at which snow can fall and the minimum temperature at which rain can fall. These two parameters can separate the input precipitation into two parts: the liquid and the iced portion,

30 and the total runoff is separated following the hydrological separation method described in Subsection 3.2. Seeing Page 5, Lines 9-20:

"Model calibration is conducted by a parallel-programmed Epsilon-Dominance Non-Dominated Sorted Genetic Algorithm II (ε-NSGA II) as proposed by the authors (Liu et al. 2017). The ε-NSGA II is coupled with Message Passing Interface (MPI) to achieve parallel autocalibration with high efficiency. As snow and frozen soil algorithms are activated, two additional

parameters related to snow modelling, namely the maximum temperature at which snow can fall ($T_{snow}$) and the minimum temperature at which rain can fall ($T_{rain}$), are optimized together with other seven conventional calibration parameters (Detailed description about the calibration of these seven typical parameters can be found in our previous studies (Liu et al., 2017). The roles of those two temperature parameters in VIC are to determine what fraction of incoming precipitation is solid

5      (snow) and liquid (rain). $T_{snow}$ and $T_{rain}$ are originally fixed for a given vegetation type. Considering glacier ablation and accumulation are simulated as snow in this study due to the absence of glacier module in the current VIC model, the ratio of solid and liquid precipitation is different from the original value. We tend to adjust them via calibration. The parameter ranges are defined as [-5,5] according to Chen et al. (2017), who used similar parameters in the CREST model for snow and glacier melting simulation."

10     **References:**

Chen, X., Long, D., Hong, Y., Zeng, C. and Yan, D.: Improved modeling of snow and glacier melting by a progressive two-stage calibration strategy with GRACE and multisource data: How snow and glacier meltwater contributes to the runoff of the Upper Brahmaputra River basin? Water Resour. Res., 53(3), 2431-2466, doi:10.1002/2016WR019656, 2017.

Dan, L., Ji, J., Xie, Z., Chen, F., Wen, G. and Richey, J.E.: Hydrological projections of climate change scenarios over the 3H

15    region of China: A VIC model assessment, J. Geophys. Res., 117(D11). doi:10.1029/2011JD017131, 2012.

Liu, L., Gao, C., Xuan, W., and Xu, Y. P.: Evaluation of medium-range ensemble flood forecasting based on calibration strategies and ensemble methods in Lanjiang Basin, Southeast China, J. Hydrol., 554, 233-250. https://doi.org/10.1016/j.jhydrol.2017.08.032, 2017.

20     **3.** The authors compare an ensemble of multiple objective Pareto simulations and single best in their ability to forecast different flood components. The authors apply Preference Ordering Routine (POR) to choose the N-Pareto-optimal sets. There is no explanation of why that particular method was used nor what is the physical meaning of the applied numerical procedure. The authors set the number of ensemble members to ten, but it is not explained why that number was chosen in statistical terms.

**Response:** The explanation for why the POR method was adopted and why the number of ten was set will be added in the

25    revised manuscript.

In the practical sense, users of automatic calibration routines have to face the task of selecting a set of suitable model parameters (preferred solution set) from the numerous Pareto-optimal sets. This is also the challenge for our study. The POR is proposed exactly to solve this kind of problem by sorting out a small number of preferred solutions. This method has been demonstrated for calibration of MIKE11/NAM rainfall-runoff model and is able to provide the best estimated parameter sets with good

30    overall model performance (Khu, 2005). These are the reasons why POR was adopted in this study.

The number of 10 is set mainly for the consideration of computational capacity. The number of parameter sets would be more than 10 when superadded the extreme value and the compromise value in any two-objective trade-off. Given the large number of ensemble meteorological forecasts and different lead times, the parameter sets less than 20 would be manageable and achievable. In the work of Khu (2005), two solutions with efficiency of order 3 with degree 4 are able to provide still good

performance, so in this study, the number of 10 is enough for representative of model parameter. As a matter of fact, when applying POR for all possible combinations of the four objective functions in this study, the final number of preferred parameter set is less than 10. Though we presupposed the upper limit, the final results didn't reach it. We have revised the paper as follows. Please see Section 3.2:

Page 6, Lines 2-7: "After calibration, a series of feasible solutions are produced by ε-NSGA II. An inevitable challenge for users of automatic calibration routines is to face the task of selecting a set of suitable model parameters (preferred solution set) from numerous Pareto-optimal sets. The method of Preference Ordering Routine (POR), developed by Khu (2005), is exactly designed to solve this kind of problem by sorting out a small number of preferred solutions. POR has been successfully applied for calibration of MIKE11/NAM rainfall-runoff model and is able to provide the best estimated parameter sets with good overall model performance. Therefore, POR is selected in this study to pick out the desired N-Pareto-optimal parameter sets…"

Page 6, Lines 15-16: "The essence of POR is to tighten the criteria of Pareto optimality, and thus enables to determine the limited preferred solutions…"

Page 6, Lines 18-23: "In this study, the POR is performed throughout all possible subspaces, and the parameter which is not dominated by any of the subspaces is retained. Additionally, some other points on the Pareto front are also retained: the extreme value for each objective function (indicated by filled circles in Fig. 2) and the compromise value in the two-objective trade-off (indicated by filled star in Fig. 2). In this way, limited number of parameter sets is picked out to represent different scenarios of model state. For convenience, the simulations driven by the N-Pareto-optimal parameter sets are referred as N-simulations, and the simulation by only one parameter set (the compromise point) is indicated by S-simulation thereafter.

…"

**4.** In the hydrograph separation subsection 3.2, the authors do not explain which data were used for the calibration/validation of the separation parameters.

**Response:** There is no parameters needed to be calibrated in Subsection 3.2. All the variables in the formulas are either known (model outputs) or unknown but able to be obtained by an iteration process like $f_{W,snow,t}$. The only related parameters are the Maximum Temperature at which snow can fall and the Minimum Temperature at which rain can fall. These two parameters determine the amount of rainfall $Rain_t$ and snowmdelt $M_t$ in Eq. (5) ($R_{snow,t} = R_t f_{R,snow,t} = R_t f_{i,snow,t} = R_t \frac{M_t}{M_t+Rain_t}$) and are calibrated with other soil parameters in VIC based on streamflow objective functions (Eq. (1)-(4)).

**5.** The post-processing of ECMWF forecasts is performed in an arbitrary way, without checking if it is necessary and provides better forecasts. Bias correction does not always give positive results regarding forecasting (Kiczko et al., 2015, Benninga et al., 2018).

**Response**: As a matter of fact, we conducted a preliminary analysis of the performance of parameterized QM on ECMWF forecasts and we found that the CRPS from bias-corrected forecasts is much smaller than that from the raw forecasts especially

for temperature forecasts. Given the already redundant figures in the original manuscript, we left them out in the Results section, but have added them in the revised manuscript as supporting information. At the same time, we added a simple conclusion in the manuscript to show readers that the post-processed forecasts are skillful enough for streamflow forecasting. Please see Page 10, Lines 21-23:

5    "Streamflow forecasts are driven by QM-SS post-processed QPF and temperature data. A preliminary analysis of raw and post-processed ECMWF forecasts reveals that QM-SS is effective for reducing errors and the post-processed forecasts are skilful enough for streamflow forecasting (seeing S.1 in supporting information)…"

[Figure]

S1. Spatial patterns of CRPS for ECMWF QPF for lead time of 3 day during wet season (May to September). (a)
10   CRPS for raw forecasts and (b) CRPS for post-processed QPF by QM-SS.

**6.** The Results section includes hydrological model performance and an assessment of flood volume and flood component forecasts. This section I find very confusing. It does not help that the authors use abbreviations that the reader needs to be acquainted with.

15   **Response**: We thought the abbreviations would be beneficial for the readers to read and understand the manuscript. According to the suggestion from referee, we reduced the abbreviations in the revised manuscript. The hydrological station names and the flood events were indicated by full name which would be helpful for reading and make less abbreviations that the reader needs to be acquainted with. At the same time, we improved our expression and wording to make the paper more readable.

20   **7**. The authors conclude that 7-day accumulated flood volumes are easier to forecast than the peak flows. The snow-induced flood component is not well captured whilst the rainfall-induced floods are forecast well. Taking into account the fact that the snow and glacier melt forecasts were not available that conclusion is not surprising.

**Response**: We agree with you that since snow and glacier melt forecasts are not available, the performance in these streamflow components is unsurprisingly inferior due to various uncertainties. We know that the snow/glacier melting is influenced by
25   not only the input rainfall and temperature but also the ability of model to capture melting process. However, considering the

rainfall-induced components (observation) are also unavailable and the portion of rainfall used to generate this component is determined by the same procedure used to determine the snowfall to generate snowmelting, the comparison between these components can tell some relatively valid conclusion. We will add the reason you suggested in the revised manuscript and it would make the conclusion more strongly justified.

**8.** The authors find that the base flow component forecast is insensitive to the forecast lead. As the base flow dynamics is slower, the forecast lead may not cover the base flow variability. However, on page 12, the authors state that for NX, the base flow forecasts show a deterioration with a lead time. A synthesis of the overall results is missing.

**Response**: We totally agree that the insensitivity of base flow to lead time is also caused by the slower flow dynamics and the lead time doesn't cover the flow variability. We have added this possible reason in the revised manuscript to make the study strongly justified. Seeing Page 12, Lines 17-20:

"Forecast skill for baseflow components seems to be insensitive to lead time (Figs. 10a-b). On one hand, the reason may be that these components are mainly generated by available water storage in the catchment. On the other hand, the baseflow process often evolves slowly, possibly making the forecast lead time not able to cover the base flow variability."

Thanks for the reminding. Due to our negligence, the unique behavior of base flow at Nuxia was missing in the conclusion. We have added the contents related to Nuxia into the final section and make the conclusion more accurate and thorough.

"At Nugesha and Yangcun stations, base flow components tend to be insensitive to increase of lead time due to the slowly-evolved baseflow process. At Nuxia Station, baseflow exhibits similar patterns to total runoff.."

**9**. The language requires correcting by a native English speaker. The text is rather difficult to follow.

**Response**: We are so sorry and the manuscript was carefully checked and polished by the native English speaker.

**Response to Editorial comments:**

**1**. Page 1: The Abstract conclusions are not transparent. It is not mentioned that the forecasting performance varies within the catchments                                                                                             studied.

**Response**: The forecasting performance does vary for different sub-catchments and the detailed difference in three sub-area has been added in the Abstract in the revised manuscript. The main difference for three sub-catchments is that baseflow components at Nuxia tends to change with the lead time. Seeing abstract:

"N-simulations is proposed to account for more scenarios of parameters in VIC. When trade-offs between multiple objectives are significant, N-simulations is recommended for better simulation and forecasting. This is why better results are obtained for Nugesha and Yangcun stations. Ensemble flood forecasting system can skilfully predict maximum floods with a lead time of more than10 days, and about 7 days ahead for melt-water related components. The accuracy of forecasts for first floods is

inferior with a lead time of only 5 days. The baseflow components for first floods are insensitive to lead time except at Nuxia Station, whilst for maximum floods an obvious deterioration in performance with lead time can be perceived."

**2**. Page 2, lines 5-9 Style should be corrected.

**Response**: We have corrected the style for lines 5-9 in Page 2.

**3**. Page 2., lines 25-27: Is this snow tracking model used in the present paper?

**Response**: Yes, this snow tracking model is used in this study.

**4**. Page 3, lines 111-13: style should be corrected.

**Response**: We have corrected the style for lines 111-113 in Page 3.

**5**. Page 5, lines 17 and 20: instead of the word "theorem" I would use "attribute".

**Response**: We have replaced the word "theorem" with "attribute" in the revised manuscript.

**6**. Pages 7- to the end: there are language problems in nearly all pages and language editing by a native English speaker is needed.

**Response**: Native English speaker was asked to check and polish the entire manuscript.

**7**. Figure 5 – there should be some quantitative assessment of the differences between simulated and observed snow cover. The comparison does not look well!

**Response**: We have added the quantitative assessment for the simulated and observed snow cover in the revised manuscript. The special correlation coefficient and the overall bias in the entire study area were calculated and attached in the figure to show more direct evaluation. Seeing Fig. 5.

[Figure]

**Figure 5. Spatial distribution of daily average snow depths derived from remote-sensing (a) and simulation by S‒simulation (b) at Nuxia.**

**8**. Figures 10-12 - it would help if the columns were named (snow-melt -induced and rainfall-induced components).

**Response**: We have named the columns in Figures 10-12 to make the figure more accessible.

[Figure]

**Figure 10. CRPSS of four different streamflow components against lead time at Nugesha. Meltwater-induced components for first floods (a) and maximum floods (c), rainfall-induced components in first floods (b) and maximum floods (d). The thick and solid lines are CRPSS by N-simulations with vertical bars showing 95% confidence intervals and the dashed lines with different markers are CRPSS by S-simulation. Black lines are meltwater/rainfall components in total runoff (Q). Red lines are CRPSS for components in surface runoff (R) and blue ones are in base flow (B).**

[Figure]

**Figure 11. CRPSS of four different streamflow components against lead time at Yangcun. Meltwater-induced components for first floods (a) and maximum floods (c), rainfall-induced components in first floods (b) and maximum floods (d). The thick and solid lines are CRPSS by N-simulations with vertical bars showing 95% confidence interval and the dashed lines with different markers are**

5   **CRPSS by S-simulation. Black lines are snowmelt/rainfall components in total runoff (Q). Red lines are CRPSS for components in surface runoff (R) and blue ones are in base flow (B).**

[Figure]

**Figure 12. CRPSS of four different streamflow components against lead time at Nuxia. Meltwater-induced components for first floods (a) and maximum floods (c), rainfall-induced components in first floods (b) and maximum floods (d). The thick and solid lines are CRPSS by N-simulations with vertical bars showing 95% confidence intervals and the dashed lines with different markers are CRPSS by S-simulation. Black lines are snowmelt/rainfall components in total runoff (Q). Red lines are CRPSS for components in surface runoff (R) and blue ones are in base flow (B).**

**9**. Benninga et al. (2018) is not referred to in the text.

**Response**: The reference for Benninga et al. (2018) was removed in the revised manuscript.

**To Referee 2**

**Response to main comments:**

**1**. The main issue in this manuscript is that the authors need more efforts to outline the advantages and disadvantages by using multiple parameter sets (N-simulations in the manuscript). It seems that more attentions are focused on the influence of multiple
5 parameter sets on entire streamflow. I would suggest the authors added more contents in discussion to clarify the merits of N-simulation for streamflow components forecasting and evaluating.

**Response**: Thank you very much for your comments. We agree that more attention should be paid to clarify the advantages and disadvantages for total runoff and component flow by using multiple parameter sets. In the original manuscript, we seem to give excessive description for entire streamflow assessment. Actually, the evaluation of multiple parameter sets on entire
10 streamflow is to verify the feasibility and applicability of N-simulations for streamflow components. As the observed component flows are unavailable, the ability in simulating entire flow to some degree indicates the ability for component flows. According to the suggestion from referee, we have discussed our results for streamflow components in more detail. Please see Subsection 4.3.

15 **2**. The main part in current manuscript is the evaluation of N-simulations for entire streamflow. In my opinion, the evaluation of entire streamflow is the foundation of evaluation of streamflow components. For there is no direct reference data for evaluation of streamflow components. The accuracy in entire streamflow is to some degree the evidence of model ability. This is the reason for usage of simulated streamflow components driven by observation as reference in components evaluation. I suggest the authors clarify this in the manuscript.

20 **Response**: Thanks for the suggestion. The evaluation of N-simulations for entire streamflow is exactly the base of evaluation for component flows, which is not mentioned in the original manuscript. To make the paper more logical, we have added this interpretation in the subsection 4.1.

Page 2, Lines 28-32: "Generally, evaluating model performance should be performed based on in-situ observations. However, observed streamflow components are usually unavailable, making the evaluation of streamflow component
25 simulations/forecasts intractable. The alternative solution is to verify forecasts on model simulations assuming that simulations driven by meteorological observations present the actual hydrological components. However, the success of this practice highly depends on how well the hydrological model is calibrated…"

Page 7, Lines 3-6: "As a result of unavailability of observed streamflow components, the evaluation of streamflow components has to be done based on the facts that the total runoff is accurately simulated and forecasted, and that the ratio of meltwater-
30 induced streamflow is similar to records in previous studies. Hence, a general assessment on total runoff simulations and forecasts is crucial and thus done first in Subsection 4.1 and 4.2…"

Page 12, Lines 5-11: "…From analysis above, an encouragingly accurate VIC simulation and forecasting system is established in YZR. This is an important precondition for subsequent evaluation of streamflow components. In some cases, for example first floods at Yangcun, VIC fails to produce accurate enough simulations and thus poor forecasts (Fig. 6b and Fig. 7c). When discussing streamflow components in these circumstances, we should evaluate model performance more carefully with consideration of errors possibly stemming from hydrological model, although the comparison of streamflow component forecasts is performed based on simulated components and to some degree, the errors in forecasts are mainly subject to meteorological inputs and therefore the hydrological error becomes negligible…"

**3**. A brief data description used for hydrological modelling should be supplemented though it is similar for the previous publication.

**Response**: As mentioned in Response to Referee #1, snow model and frozen algorithm are used in the current study, which is different to the previous study. In this way, the description of related snow depth data and additional calibration parameters were added in Subsection 2.2 and 3.1 respectively.

Page 4, Lines 18-23: "Snow depth data provided by Cold and Arid Regions Science Data Center at Lanzhou, China (http://westdc.westgis.ac.cn/) are used to evaluate snow depth simulations. The data is derived from passive microwave remote sensing at a resolution of $0.25° \times 0.25°$ (Che et al., 2008; Dai et al., 2015). The digital elevation model (DEM) data used in the hydrological model is downloaded from Geospatial Data Cloud (http://www.gscloud.cn) at the resolution of 90 m×90 m. The vegetation and soil parameters in the model are defined according to 1 km China soil map based on Harmonized World Soil Database (Fischer et al., 2008) and 1 km land cover products of China (Ran et al., 2010)."

Page 5, Lines 11-20: As snow and frozen soil algorithms are activated, two additional parameters related to snow modelling, namely the maximum temperature at which snow can fall ($T_{snow}$) and the minimum temperature at which rain can fall ($T_{rain}$), are optimized together with other seven conventional calibration parameters (Detailed description about the calibration of these seven typical parameters can be found in our previous studies (Liu et al., 2017). The roles of those two temperature parameters in VIC are to determine what fraction of incoming precipitation is solid (snow) and liquid (rain). $T_{snow}$ and $T_{rain}$ are originally fixed for a given vegetation type. Considering glacier ablation and accumulation are simulated as snow in this study due to the absence of glacier module in the current VIC model, the ratio of solid and liquid precipitation is different from the original value. We tend to adjust them via calibration. The parameter ranges are defined as [-5,5] according to Chen et al. (2017), who used similar parameters in the CREST model for snow and glacier melting simulation.

[Figure]

20    **Figure 7.** $E_q$ **of first floods for Q1 and Q7 at (a)-(b) Nugesha, (c)-(d) Yangcun, and (e)-(f) Nuxia. The black-coloured boxplots are forecasts driven by S-simulation and forecasts derived from N-simulations are denoted by grey.**

[Figure]

**Figure 8.** $E_q$ of maximum floods for Q1 and Q7 at (a)-(b) Nugesha, (c)-(d) Yangcun, and (e)-(f) Nuxia. The black-coloured boxplots are forecasts driven by S-simulation and forecasts derived from N-simulations are denoted by grey.

[Figure]

5  **Figure 9.** $E_t$ for first flood and maximum flood at (a)-(b) Nugesha, (c)-(d) Yangcun, and (e)-(f) Nuxia. The black-coloured boxplots are forecasts driven by S-simulation and forecasts derived from N-simulations are denoted by grey.

**6**. I could understand most of the manuscript without difficulty, and the methods are well documented. However, there are many minor errors in spelling, grammar and English style that I have not corrected. I recommend that the authors have the manuscript proof-read by a native English speaker before publication.

**Response**: The manuscript has been carefully checked and polished by a native English speaker.

**Response to specific comments:**

**7**. Page 6, Line 4: as snow and glacier melt water is considered together, term "meltwater" is better representative than "snowmelt".

**Response**: The term "snowmelt" has been replaced by "meltwater" in the revised manuscript.

**8**. Page 9, Line 31: "Lead times of 3, 5, 7, 10, 12 and 14 days" make it "Lead times of day 3, 5, 7, 10, 12 and 14"

**Response**: "Lead times of 3,5,7,10,14 days" has been changed into "Lead times of day 3, 5,7,10,12,14".

**9**. Page 9, Line 32-33: "Generally, flood volumes tend to be better captured with the increase of "duration", especially for lead times from 7 day to 12 day" should be explained.

**Response**: The detailed explanation was added in the revised manuscript. Seeing Page 10, Lines 25-27:

"Generally, flood volumes tend to be better captured with the increase of duration. One reason is that there are often larger errors in simulated flood peak, making the single day flood volume more prone to bias. Another reason is that when the duration increases, the bias from streamflow for this relatively long period can offset with each other."

**10**. Page 10, Line 32: "It seems that N-simulations scheme works in poorly-calibrated regions." from what results the authors draw this conclusion.

**Response**: When we compared Figures 7-9, we found that Nuxia is the only case that N-simulations consistently provide comparable or even better results than S-simulation when verified on simulated streamflow. Since the contents related to evaluation on simulated streamflow has been left out, this sentence was deleted in the revised manuscript.

**11**. Page 13, Lines 2: "but" make it "while"

**Response**: "but" is changed into "while".

**12**. Page 13, Lines 27: "We believe that the phenomenon captured by most of the parameter sets would be the most possible truth." Change it to "From the view of ensemble, the phenomenon captured by more parameter sets is regarded as the most possible occurrence."

[revised manuscript text omitted]

---

## Referee Report (RR1)

Major comments have been well addressed by the author, there are just a few minor issues need to be modified.

Page 9 line 10: the outperformance during evaluation period could also be related to the shorter length of the timespan.

Page 9 line 15: please add some references to support the opinion.

Page 9 line 31: "for the output snow depth from VIC is actually the sum of snow and glacier/ice" is redundant, delete it.

Page 13 line 31: change "compared" into "comparing".

Page 14 line 10: since you mentioned other two methods calculating the glacier melt, and meanwhile in line 13 stated "Overly complicated methods probably bring out more uncertainties….more observations are available with the development of technologies in the future, more elaborate separation method is expected" what's the observed input for these two models, are they overly complicated and with more uncertainty? If not, please modify the deduction since it cannot be reached either based on your study or the studies you referred to.

In the discussion section, please add some references on the similar studies using the proxy method and make a comparison with this study.

Page 18, line 25:  please add the year for the reference.

---

## Author Response (AR2)

**Revision notes for "Potential application of hydrological ensemble prediction in forecasting flood and its components over the Yarlung Zangbo River Basin, China" (hess-2018-179)**

Dear Editor and referees,

Thanks a lot for your great efforts to review this manuscript and give very valuable comments. We agree with your suggestions which will be of great help to improve the quality of our manuscript. Here we have addressed the comments from you and the detailed description is attached in this document.

10  Best regards,

Li Liu, Yue-Ping Xu,Suli Pan, Zhixu Bai

**To Referee #2**

1. Page 9 line 10: the out-performance during evaluation period could also be related to the shorter length of the timespan.

15  **Response**: Thanks for your suggestion. We have added this reason in the revised manuscript.

"Generally speaking, the model performance during evaluation is more satisfactory than that during calibration. It is probably caused by the existence of considerable extraordinary flood events during the calibration period. The relatively shorter timespan in evaluation is also one of the reasons."

20  2. Page 9 line 15: please add some references to support the opinion.

**Response**: We have added some previous studies to justify the opinion. Please see Page 9, Line 15-17:

"We also guess that within downstream regions the hydrological process becomes too complicated due to human activities to be simulated by models (Li et al., 2013; Liu et al., 2014)."

Li, F., Xu, Z., Feng, Y., Liu, M. and Liu, W.: Changes of land cover in the Yarlung Tsangpo River basin from 1985 to 2005,

25  Environ. Earth Sci., 68, 181-188, ,2013.

Liu, Z., Yao, Z., Huang, H., Wu, S., and Liu, G.: Land use and climate changes and their impacts on runoff in the Yarlung Zangbo river basin, China, Land Degrad. Dev., 25, 203-215, doi:10.1002/ldr.1159, 2014.

3. Page 9 line 31: "for the output snow depth from VIC is actually the sum of snow and glacier/ice" is redundant, delete it.

30  **Response**: We have deleted this sentence in the revised manuscript.

4. Page 13 line 31: change "compared" into "comparing".

**Response**: We have changed "compared" with "comparing".

5. Page 14 line 10: since you mentioned other two methods calculating the glacier melt, and meanwhile in line 13 stated "Overly complicated methods probably bring out more uncertainties….more observations are available with the development of technologies in the future, more elaborate separation method is expected" what's the observed input for these two models, are they overly complicated and with more uncertainty? If not, please modify the deduction since it cannot be reached either based on your study or the studies you referred to.

In the discussion section, please add some references on the similar studies using the proxy method and make a comparison with this study.

**Response**: Thank you for your suggestion.

(1) What's the observed input for these two models, are they overly complicated and with more uncertainty? If not, please modify the deduction since it cannot be reached either based on your study or the studies you referred to.

The simple degree-day glacier algorithm requires only the temperature data and the distribution of glacier in study area. The significant defect of this method is without considering water balance. The energy balance method requires incoming longwave radiation, emitted longwave radiation, outgoing longwave radiation and other data to calculate related glacier surface-energy balance and mass balance. If all those data are calculated from air temperature rather than observation, the sparse distributed meteorological network is highly likely to bring in additional uncertainties.

The deduction here is somewhat deviated from our conclusion, and we have modified the related part based on our study.

(2) In the discussion section, please add some references on the similar studies using the proxy method and make a comparison with this study.

The practice using proxy as observations is common when observed streamflow is absent. Similar studies can be found in Arnal et al. (2018) and Harrigan et al. (2018). We have added the relevant studies in the discussion section.

"For streamflow components forecast, the biggest challenge is the absence of data series of in-situ streamflow components. Therefore, in this study the simulation driven by observed forcing becomes an alternative to act as proxy and thus the error stemming from hydrological model is avoided. This is a common practice when observation is absent (Arnal et al., 2018; Harrigan et al., 2018)."

Arnal, L., Cloke, H.L., Stephens, E., Wetterhall, F., Prudhomme, C., Neumann, J., Krzeminski, B. and Pappenberger, F.,: Skilful seasonal forecasts of streamflow over Europe? Hydrol. Earth Syst. Sci., 22(4), 2057-2072, 2018.

Harrigan, S., Prudhomme, C., Parry, S., Smith, K. and Tanguy, M.: Benchmarking ensemble streamflow prediction skill in the UK, Hydrol. Earth Syst. Sci., 22(3), 2023-2039, 2018.

Page 18, line 25: please add the year for the reference.
**Response**: We have added the year for the reference.

**To Referee #3**

1. How the "ability" of ensemble predictions to forecast can be defined? It could be expected that some weather projections predict observed flood peak and other not. At which point the hypothesis can be falsified? What is a level of CRPS, CRPSS or MAE at which authors would agree that ensemble predictions are unable to forecast a flood peak? In my opinion it always is
5  possible to insist that the approach is "able" - better or worse.
The problem should be rather analysed in terms of suitability, where the ensemble predictions are compared with other methods. Then it would be possible quantifying the performance of different methods using i.e. CRPSS. Such an approach can be found i.e. in Pappenberger et al. (2005) who analyzed ensemble along with deterministic predictions.
**Response**: Thank you very much.
10  (1) We agree that it is always difficult to interpret CRPS, CRPSS and MAE. However, it is still the most commonly used metrics to quantify ensemble forecasts. We have read the recommended study by Pappenberger et al. (2005). It is a good example to analyze ensemble with deterministic predictions, but, in our opinion, it is not suitable for our study. Firstly, in the study of Pappenberger et al. (2005) (hereinafter referred as "Pappenberger' study"), only one case study is used, and thus the plot of hydrograph against lead time is feasible. In our study, we choose more than ten case studies for each station, making it
15  redundant to adopt similar plots. Secondly, the deterministic predictions in Pappenberger' study consisted of 6 ensembles, and it is convenient to graph all the hydrographs, but in our study the S-simulation prediction is still composed by 51 ensembles.
(2) We totally agree to classify the problem as suitability, and we have revised the expression in the manuscript. The CRPSS has been used to quantify the model performance in original paper as shown in Fig. 6 and Fig. 10-13.
"The two purposes of this study are therefore to investigate the suitability of HEPS in forecasting flood volume and its
20  components over cold and mountainous area, and the impact of an ensemble of selected pareto optimal solutions on model simulation and forecasting compared to a single parameter set."

2. Hydrograph separation is probably the weakest part of the study. According to the text, glaciers are important part of the basin system. However, the applied model does not account for this component. In Page 5, line 16-18 authors assume that it
25  can be described as snow accumulation. This is a very rough assumption and its limitations can be seen in Fig 3 (authors are aware of this fact, page 8, lines 25-26), where flows, probably shaped by glacier outflow, are not explained. What is the point analysing forecasted streamflow components section 4.3), if the essential element of the base flow is missing? The conclusion to this remark could be the sentence from the Discussion section, referring to the forecasted components (Page 14, lines 18-19):
30  "Therefore, in this study the simulation driven by observations becomes an alternative to act as proxy although it is difficult to determine whether such proxy is believable or not." If it is hard to determine that we can believe the method or not, it is not a scientific approach.

**Response**: It is our mistake to unclearly describe the glacier part. Actually, the glacier only takes up about 2% of the area in the YZR (Zhang et al., 2013), and of streamflow less than 10% (Chen et al., 2017). The underestimation of baseflow in Fig.3 is not, to a great degree, caused by lack of glacier modeling. For the low flow period is the time when the glacier should be accumulated, the absence of glacier is supposed to overestimate baseflows. We have compared our results with a previous

5  study (Zhang et al., 2013, Fig.3(e), the study period is somewhat different, from 1961 to 1999 in that study), similar underestimation exits in low flow period, even though a glacier is embedded in the hydrological model. The authors thought the errors were mainly from the observed inputs. However, how much the inputs might be underestimated for the YZR is actually unknown. In this study, one of the reasons for this phenomenon is that the objective functions used to calibrate hydrological model emphasize more on high flows. The underestimation is, in the meanwhile, caused by the errors of

10  meteorological measurements in the study area, which has been documented by previous studies like Tong et al. (2014) and Zhang et al. (2013).

[Figure]

Fig.11 in Tong et al. (2014)

[Figure]

**Figure 3.** Mean monthly simulated and observed streamflow for the six source river basins in the TP. The data periods are the same as in Figure 2.

15                                           Fig.3 in Zhang et al. (2013)

Second, the snow is truly an important contributor to total runoff, especially during spring. It is reported that near 25% of the total runoff is derived from snowmelt water (Zhang et al., 2013). Most of previous studies (Li et al., 2014; Liu et al., 2014; Sun et al., 2013) considered glacier together with snow when simulating streamflow components in this study area, as the

glacier is neglectable if compared to snow. Another reason to modelling glacier as snow is that based on current hydrological model structure and available observations, it is hard to separate glacier from snow. Finally, our study is more interested in meltwater induced streamflow rather than snow and glacier meltwater streamflow respectively.

As for "What is the point analyzing forecasted streamflow components section 4.3", it is because the model simulation is
5 possibly biased and using simulation driven by observed forcing data as observation proxy is an effective way to remove hydrological errors and is suitable to compare the N-simulations and S-simulation in forecasts components.

All of abovementioned contents has been added or rewrote in the revised manuscript.

"The observed and simulated hydrographs during the evaluation period at Nuxia are presented in Fig. 3. An obvious
10 underestimation can be observed in low flow periods, which is similar to previous studies by Zhang et al. (2013) and Tong et al. (2014). The absence of glacier module in VIC is believed to has limited influence on this underestimation, for similarly underestimated low flow was found when glacier modelling was embedded in VIC (Zhang et al., 2013). For our study, the underestimation is, in the meanwhile, caused by the fact that the objective functions used for calibration have the tendency to give more attention to high flows, as the flood is the focus of our investigation. As noticed in Fig. 3, the flood peaks are well
15 captured by S-simulation in most cases. N-simulations are able to cover all the extreme values while sometimes slight overestimation exists."

Li, F., Xu, Z., Liu, W. and Zhang, Y.: The impact of climate change on runoff in the Yarlung Tsangpo River basin in the Tibetan Plateau. Stoch. Env. Res. Risk A., 28(3), 517-526, 2014

Liu, Z., Yao, Z., Huang, H., Wu, S. and Liu, G.: Land use and climate changes and their impacts on runoff in the Yarlung Zangbo river basin,
20 Land Degrad. Dev., 25(3), 203-215, DOI: 10.1002/ldr.1159, 2014

Chen, X., Long, D., Hong, Y., Zeng, C. and Yan, D.: Improved modeling of snow and glacier melting by a progressive two-stage calibration strategy with GRACE and multisource data: How snow and glacier meltwater contributes to the runoff of the Upper Brahmaputra River basin? Water Resour. Res., 53, 2431-2466, doi:10.1002/2016WR019656, 2017.

Tong, K., Su, F., Yang, D., Zhang, L., and Hao, Z.: Tibetan Plateau precipitation as depicted by gauge observations, reanalyses and satellite
25 retrievals, Int. J. Remote Sens., 34, 265-285, doi:10.1002/joc.3682, 2014.

Sun, R., Zhang, X., Sun, Y., Zheng, D. and Fraedrich, K.: SWAT-based streamflow estimation and its responses to climate change in the Kadongjia River watershed, southern Tibet, J. Hydrometeorol., 14(5), 1571-1586, 2013

Zhang, L., Su, F., Yang, D., Hao, Z., and Tong, K.: Discharge regime and simulation for the upstream of major rivers over Tibetan Plateau, J. Geophys. Res., 118, 8500-8518, doi:10.1002/jgrd.50665, 2013.

3) The concept of the ensemble of Pareto optimal parameters is the most interesting, but not addressed properly. Authors have identified a N-set of parameters that provides the tradeoffs between four fit measures: the Nash–Sutcliffe efficiency and relative bias for whole flow time series and limited to 10% of highest flows. However, the performance of the forecast system is analyzed only in the respect of a single measure: accumulated flood volumes. For such a single criterion it is quite obvious

that a single compromise simulation should be better. The advantage of the parameter ensemble could be found, if other measures, similar to those in calibration, were also included.

**Response**: Thanks for the referee being interested in our study. We agree that accumulated flood volumes only present the function of 10% of highest flows, but this sentence is right only when we are talking about the annual maximum flood. For annual first flood (spring flood) and the peak time simulating, it is more related to the whole hydrograph. This means actually both flood volumes and peak time are considered in evaluating the performance of the forecast system.

4) In the reviewer's opinion, authors should revise their research design, focusing maybe on the forecast quality, in the respect of identified parameters (Pareto-sets), instead of stream flow components.

**Response**: As a matter of fact, the focus of this study is both forecasts in total runoff and streamflow components. This is the reason why we constructed the manuscript as the order of "streamflow simulating-total runoff forecasting-streamflow components forecasting". We found the problem in our previous study which made the streamflow components forecasting an ultimate goal. We have revised the structure of the manuscript. Please check the revised submission.

A) The description of Snowmelt Tracking Algorithm, provided in Section 3.3 is unclear and not supported with references to Li et al. (2017). The original Snowmelt Tracking Algorithm Li et al. 2017) acquires internal variables of a runoff model:

"In this study we develop explicit quantification of the historical and future fQ , snow over the western U.S. We quantify fQ , snow by tracking the fate of snowmelt in modeled hydrologic fluxes" - Li et al. (2017).

In my opinion all assumptions used in eq. (5-7) should be justified and explained by authors. It is unclear, why it is possible to assume, that infiltration (f_{i,snow}) and runoff ratios (f_{R,snow}) are equal (Eq. 5). Also Eq. 7 is unclear. What is a definition of fi,snow? Equiation 5 suggests that it is a ratio: fi,snow = \frac{I_snow}{R}. Why in Eq. 7 it is multiplied by i (infiltration?)? The same applies to F_{w, snow}.

Please, provide also dimensions for variables: R, B, W and ETt.

**Response**: (1) Thank you very much for this comment. We don't assume the infiltration and runoff ratios to be equal. As a matter of fact, we only assume meltwater and rainfall exhibit identical infiltration and runoff ratios. But we did make some mistakes when writing Eq. (5) in the original manuscript. We have checked our note and code, and make sure that this is just a writing error. We are so sorry to overlook this mistake in the last revision.

(2) The definition of $f_{i,snow,t}$ is the ratio of snow-induced infiltration in total infiltration. Thus, it can be multiplied by total infiltration ($i_t$).

(3) The dimensions for variables $R_t$, $B_t$, $W_t$ and $ET_t$ are millimeter and have been added in the manuscript. We show the results at the basin average.

B) Page 3, Lines 10-15 are unclear and please consider rewriting. Parameters ensembles of a hydrological model and parameter sets refer here to a wider problem of the hydrological model uncertainty. Mentioned techniques are just possible solutions. I suggest defining this problem as an uncertainty analysis and provided these references as examples. Please note, that in the present form the text is confusing, as it is hard to distinguish between "forecast ensemble" (meteorological model uncertainty) and "parameter ensemble" (for hydrological models' uncertainty).

**Response**: Thank you very much for your suggestion. We are sorry to mix the concept of meteorological uncertainty and hydrological uncertainty. In the revised manuscript, we have defined the uncertainty from parameter of hydrological model as hydrological uncertainty and added the relevant references. Please see Page 3, Lines 10-15:

"Due to this limitation, utilizing an ensemble of parameter sets to represent uncertainty from hydrological model is referential. Pappenberger et al. (2005) used six different parameter sets to identify uncertainty from hydrological model. Teutschbein and Seibert (2012) employed 100 different optimized parameter sets in HBV to simulate streamflow in order to consider parameter uncertainty. The basic principle in ensemble forecasts is using ensemble spread to quantify forecast uncertainty and thus provide essential information to users (Bauer et al., 2015). Analogous to this concept, the benefit of adopting an ensemble of parameter sets from Pareto optimal front by multi-objective optimization algorithm for flood forecasting with consideration of hydrological parameter uncertainty remains unresolved and is noteworthy to investigate. Especially for streamflow components modelling and forecasting, with limited or unavailable observations, it is impossible to achieve rigorous calibration, and thus accounting for hydrological parameter uncertainty is necessary (Pappenberger et al., 2005)."

Pappenberger, F., Beven, K.J., Hunter, N.M., Bates, P.D., Gouweleeuw, B.T., Thielen, J. and De Roo, A.P.J., Cascading model uncertainty from medium range weather forecasts (10 days) through a rainfall-runoff model to flood inundation predictions within the European Flood Forecasting System (EFFS), Hydrol. Earth Syst. Sci., 9(4), 381-393, 2005.

C) Page 4, Lines 16-17: Please note, that the supporting references are more than 4 years old.

**Response**: Thanks for your reminder. We have replaced those supporting references with the latest ones. Please see Page 4, Lines:

"ECMWF is selected in this study due to the well-known fact that forecasts from ECMWF are more skilful than other Ensemble Prediction Systems in TIGGE database (Aminyavari et al., 2018; Louvet et al., 2016; Hamill and Scheuerer, 2018)."

Louvet, S., Sultan, B., Janicot, S., Kamsu-Tamo, P.H. and Ndiaye, O.: Evaluation of TIGGE precipitation forecasts over West Africa at intraseasonal time scale,Clim.Dynam., 47(1-2), 31-47, DOI 10.1007/s00382-015-2820-x, 2016.

Aminyavari, S., Saghafian, B. and Delavar, M.: Evaluation of TIGGE Ensemble Forecasts of Precipitation in Distinct Climate Regions in Iran, Adv. Atmos. Sci., 35(4), 457-468, 2018.

Hamill, T.M. and Scheuerer, M.: Probabilistic Precipitation Forecast Postprocessing Using Quantile Mapping and Rank-Weighted Best-Member Dressing, Mon. Weather Rev., 146(12), 4079-4098, 2018.

D) Page 5, Line 7, what data was used in cross-validation - from these 27 meteorological stations? If so, it should not be written, that lapse rates were estimated using available meteorological data?

**Response**: Yes, records from these 27 meteorological stations were used in a leave-one-out-cross-validation. The lapse rate is defined by performing the least square fitting considering elevation.

E) Section 4.2. It is unclear for how a flood forecasting system was tested. Did authors use a receding horizon or forecast was issued for chosen events? Terms "first floods" maximum floods etc. should be defined.

**Response**: We didn't use the receding horizon. The forecast was issued for each chosen event. We have added the relevant information in the revised manuscript. The terms of "first floods" and "maximum floods" were defined in the original
10   manuscript in Section 3.5.

"The annual maximum flood is picked out as typical flood events. Meanwhile, the first flood event in each year is also selected. Maximum flood is determined by the maximum daily streamflow in a year. For first flood, the definition seems to be slightly subjective. Nevertheless, first flood is just introduced as an example to verify the skill of VIC/ECMWF system to forecast the meltwater components. There are three criterions for us to define the first flood: (1) the peak flow should be more than twice
15   of the average daily streamflow during dry period (November to March); (2) the duration for the flood event should be longer than 7 days; (3) the observed snowpack is present. Forecasts were issued for each chosen event. Considering that maximum flood events in YZR usually last for several months, flood volume over the entire flood event is impossible to be covered by medium-range weather forecasts. Four typical flood volumes are therefore chosen to represent the volume performance…"

20   F) Page 13, Line 21: "Firstly, N-simulations generally behave better when the trade-offs in multi-objectives are significant." does the study supports this finding?

**Response**: This conclusion was drawn based on the results that N-simulations generally behave poor at Nuxia station, while for Nugesha and Yangcun N-simulations are better than S-simulation (Table 1 and 2). The difference for these three stations is that the trade-off between two objective functions is arc-shaped at Nugesha and Yangcun stations, indicating when moving
25   on the front the improvement in one objective function will result in deterioration in the other objective function. While at Nuxia station, the compromise point is almost the best parameter set with highest score in most of the objective functions. From this perspective, we think that N-simulations generally behave better when the trade-offs in multi-objectives are significant. We possibly didn't explain this clearly in the previous study, and we have added this in the revised manuscript. Please see Section 4.1 and Conclusion:

[revised manuscript text omitted]